# MuseGNN: Forming Scalable, Convergent GNN Layers that Minimize a Sampling-Based Energy

**Haitian Jiang**[1]* **Renjie Liu**[2] **Zengfeng Hunag**[3] **Yichuan Wang**[4]

**Xiao Yan**[5] **Zhenkun Cai**[6] **Minjie Wang**[6] **David Wipf**[6]†

[1]New York University [2]Southern University of Science and Technology [3]Fudan University
[4]UC Berkeley [5]Centre for Perceptual and Interactive Intelligence [6]Amazon Web Services

*haitian.jiang@nyu.edu †davidwipf@gmail.com

## Abstract

Among the many variants of graph neural network (GNN) architectures capable of modeling data with cross-instance relations, an important subclass involves layers designed such that the forward pass iteratively reduces a graph-regularized energy function of interest. In this way, node embeddings produced at the output layer dually serve as both predictive features for solving downstream tasks (e.g., node classification) and energy function minimizers that inherit transparent, exploitable inductive biases and interpretability. However, scaling GNN architectures constructed in this way remains challenging, in part because the convergence of the forward pass may involve models with considerable depth. To tackle this limitation, we propose a sampling-based energy function and scalable GNN layers that iteratively reduce it, guided by convergence guarantees in certain settings. We also instantiate a full GNN architecture based on these designs, and the model achieves competitive accuracy and scalability when applied to the largest publicly-available node classification benchmark exceeding 1TB in size. Our source code is available at https://github.com/haitian-jiang/MuseGNN.

## 1 Introduction

Graph neural networks (GNNs) are a powerful class of deep learning models designed specifically for graph-structured data. Unlike conventional neural networks that primarily operate on independent samples, GNNs excel in capturing the complex cross-instance relations modeled by graphs (Hamilton et al., 2017; Kearnes et al., 2016; Kipf & Welling, 2017; Velickovic et al., 2018). Foundational to GNNs is the notion of message passing (Gilmer et al., 2017), whereby nodes iteratively gather information from neighbors to update their representations. In doing so, information can propagate across the graph in the form of node embeddings, reflecting both local patterns and global network effects, which are required by downstream tasks such as node classification.

Among many GNN architectures, one promising subclass is based on graph propagation layers derived to be in a one-to-one correspondence with the descent iterations of a graph-regularized energy function (Ahn et al., 2022; Chen et al., 2022a; 2021; Yang et al., 2021; Ma et al., 2021; Gasteiger et al., 2019; Pan et al., 2020; Zhang et al., 2020; Zhu et al., 2021; Xue et al., 2023). For these models, the layers of the GNN forward pass computes increasingly-refined approximations to a minimizer of the aforementioned energy function. Importantly, if such energy minimizers possess interpretable properties or inductive biases, the corresponding GNN architecture naturally inherits them (Zheng et al., 2024), unlike traditional GNN constructions that may be less transparent. More broadly, this association between the GNN forward pass and optimization can be exploited to introduce targeted architectural enhancements (e.g., robustly handling graphs with spurious edges and/or heterophily (Fu et al., 2023; Yang et al., 2021). Borrowing from Yang et al. (2022), we will refer to models of

---

*Work partially completed during an internship at Amazon Web Services.
†Corresponding author.

this genre as *unfolded* GNNs, or UGNNs for short, given that the layers are derived from a so-called unfolded (in time/iteration) energy descent process.

Despite their merits w.r.t. interpretability, UGNNs face non-trivial scalability challenges. This is in part because they can be constructed with arbitrary depth (i.e., arbitrary descent iterations) while still avoiding undesirable oversmoothing effects (Oono & Suzuki, 2020; Li et al., 2018), and the computational cost and memory requirements of this flexibility are often prohibitively high. Even so, there exists limited prior work explicitly tackling the scalability of UGNNs, or providing any complementary guarantees regarding convergence on large graphs. Hence most UGNN models are presently evaluated on relatively small benchmarks.

To address these shortcomings, we propose a scalable UGNN model that incorporates efficient subgraph sampling within the fabric of the requisite energy function. Our design of this model is guided by the following three desiderata: (i) maintain the characteristic properties, interpretability, and extensible structure of full-graph unfolded GNN models; (ii) using a consistent core architecture, preserve competitive accuracy across datasets of varying size, *including the very largest publicly-available benchmarks*; and (iii) do not introduce undue computational or memory overhead beyond what is required to train the most common GNN alternatives. Our proposed model, which we will later demonstrate satisfies each of the above, is termed *MuseGNN* in reference to a GNN architecture produced by the minimization of an unfolded sampling-based energy. Building on background motivations presented in Section 2, our MuseGNN-related contributions are three-fold:

1. In Sections 3 and 4, we expand a widely-used UGNN framework to incorporate offline sampling into the architecture-inducing energy function design itself, as opposed to a post hoc application of sampling to existing GNN methods. The resulting model, which we term MuseGNN, allows us to combine the attractive properties of UGNNs with the scalability of canonical GNNs. Notably, by design MuseGNN can exploit an unbiased estimator of the original full-graph energy if desired, or else sampling-based alternatives that are provably more expressive.

2. We prove in Section 5 that MuseGNN possesses desirable convergence properties regarding both upper-level (traditional model training) and lower-level (interpretable energy function descent) optimization processes. A supporting empirical example of the latter is illustrated in Figure 1 above. Critically, our convergence gaurantees are agnostic to the particular offline sampling operator that is adopted, and so *any* future options can be seamlessly integrated. These attributes increase our confidence in reliable performance when moving to new problem domains that may deviate from known benchmarks.

3. Finally, in Section 6 we provide complementary empirical support that MuseGNN performance is stable in practice, preserving competitive accuracy and scalability across task size. En route, we achieve SOTA performance w.r.t. homogeneous graph models applied to the largest, publicly-available node classification datasets from OGB and IGB exceeding 1TB in size.

Table 1 contextualizes MuseGNN w.r.t. prior work, with details to follow in subsequent sections. Specifically, we compare with those commonly-used GNN architectures that have been successfully scaled and validated on the largest public graphs using some form of neighbor sampling (NS) (Hamilton et al., 2017; Huang et al., 2024; Waleffe et al., 2023), or alternatively, GNNAutoScale (GAS) (Fey et al., 2021; Yu et al., 2022). Note that most GNNs populating competitive leaderboards such as OGB *do not presently scale to the largest graphs*, nor do they possess properties of UGNNs, and hence are not our focus. And finally, we compare with existing full-graph (FG) UGNN approaches, as well as USP (Li et al., 2022) and LazyGNN (Xue et al., 2023), two recent frameworks explicitly designed for scaling UGNNs. Appendix B.1 also considers additional scalable GNN models.

## 2 BACKGROUND AND MOTIVATION

### 2.1 GNN ARCHITECTURES FROM UNFOLDED OPTIMIZATION

**Notation.** Let $\mathcal{G} = \{\mathcal{V}, \mathcal{E}\}$ denote a graph with $n = |\mathcal{V}|$ nodes and edge set $\mathcal{E}$. We define $D$ and $A$ as the degree and adjacency matrices of $\mathcal{G}$ such that the corresponding graph Laplacian is $L = D - A$. Furthermore, associated with each node is both a $d$-dimensional feature vector, and a $d'$-dimensional label vector, the respective collections of which are given by $X \in \mathbb{R}^{n \times d}$ and $T \in \mathbb{R}^{n \times d'}$.

Table 1: MuseGNN vs. existing methods on largest graphs (LG), where 'top acc.' refers to top LG accuracy. Note that the convergence guarantee and greater energy expressivity are specifically defined w.r.t. UGNN models, hence 'N/A' for non-UGNNs without a lower-level energy.

| | energy descent | converge guarantee | handle LGs | top acc. | ↑ energy expressivity |
|---|---|---|---|---|---|
| GNN+NS | ✗ | N/A | ✓ | ✗ | N/A |
| GNN+GAS | ✗ | N/A | ✗ | ✗ | N/A |
| UGNN(FG) | ✓ | ✓ | ✗ | ✗ | ✗ |
| USP | ✓ | partial | ✗ | ✗ | ✗ |
| LazyGNN | ✓ | ✗ | ✗ | ✗ | ✗ |
| MuseGNN | ✓ | ✓ | ✓ | ✓ | ✓ |

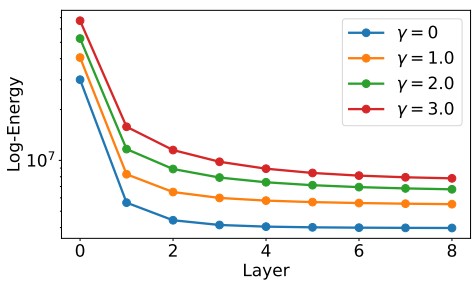

Figure 1: MuseGNN *forward pass* convergence, energy descent on `ogbn-papers100M` dataset with varying $\gamma$ (to be introduced in Section 3.2).

**GNN Basics.** Given a graph defined as above, canonical GNN architectures are designed to produce a sequence of node-wise embeddings $\{Y^{(k)}\}_{k=1}^{K}$ that are increasingly refined across $K$ model layers according to the rule $Y^{(k)} = h(Y^{(k-1)}; W, A, X) \in \mathbb{R}^{n \times d}$. Here $h$ represents a function that updates $Y^{(k-1)}$ based on trainable model weights $W$ as well as $A$ (graph structure) and optionally $X$ (input node features). To facilitate downstream tasks such as node classification, $W$ may be trained, along with additional parameters $\theta$ of any application-specific output layer $g : \mathbb{R}^d \to \mathbb{R}^{d'}$, to minimize a loss of the form

$$\mathcal{L}(W, \theta) = \sum_{i=1}^{n'} \mathcal{D}\left( g\left[ Y^{(K)}(W)_i ; \theta \right], T_i \right). \tag{1}$$

In this expression, $Y_i^{(K)} \equiv Y^{(K)}(W)_i$ reflects the explicit dependency of GNN embeddings on $W$ and the subscript $i$ denotes the $i$-th row of a matrix. Additionally, $n'$ refers to the number of nodes in $\mathcal{G}$ available for training, while $\mathcal{D}$ is a discriminator function such as cross-entropy. We will sometimes refer to the training loss (1) as an *upper-level* energy function to differentiate its role from the lower-level energy defined next.

**Moving to Unfolded GNNs.** An *unfolded* GNN architecture ensues when the functional form of $h$ is explicitly chosen to align with the update rules that minimize a second, *lower-level* energy function denoted as $\ell(Y)$. In this way, it follows that $\ell\left(Y^{(k)}\right) = \ell\left(h\left[Y^{(k-1)}; W, A, X\right]\right) \leq \ell\left(Y^{(k-1)}\right)$, This restriction on $h$ is imposed to introduce desirable inductive biases on the resulting GNN layers that stem from properties of the energy $\ell$ and its corresponding minimizers (see Section 2.2 ).

While there are many possible domain-specific choices for $\ell(Y)$ in the growing literature on unfolded GNNs (Ahn et al., 2022; Chen et al., 2022a; Ma et al., 2021; Gasteiger et al., 2019; Pan et al., 2020; Yang et al., 2021; Zhang et al., 2020; Zhu et al., 2021; Xue et al., 2023), we will focus our attention on a particular form that encompasses many existing works as special cases, and can be easily generalized to cover many others. Originally inspired by Zhou et al. (2003) and generalized by Yang et al. (2021) to account for node-wise nonlinearities, we consider

$$\ell(Y) := \|Y - f(X; W)\|_F^2 + \lambda \operatorname{tr}(Y^\top L Y) + \sum_{i=1}^{n} \zeta(Y_i), \tag{2}$$

where $\lambda > 0$ is a trade-off parameter, $f$ represents a trainable base model parameterized by $W$ (e.g., a linear layer or MLP), and the function $\zeta$ denotes a (possibly non-smooth) penalty on individual node embeddings. The first term in (2) favors embeddings that resemble the input features as processed by the base model, while the second encourages smoothness across graph edges. And finally, the last term is included to enforce additional node-wise constraints (e.g., non-negativity).

We now examine the form of $h$ that can be induced when we optimize (2). Although $\zeta$ may be non-smooth and incompatible with vanilla gradient descent, we can nonetheless apply proximal gradient descent in such circumstances (Combettes & Pesquet, 2011), leading to the descent step

$$Y^{(k)} = h\left(Y^{(k-1)}; W, A, X\right) = \operatorname{prox}_\zeta\left(Y^{(k-1)} - \alpha\left[(I + \lambda L)Y^{(k-1)} - f(X; W)\right]\right), \tag{3}$$

where $\alpha$ controls the learning rate and $\operatorname{prox}_\zeta(u) := \arg\min_y \frac{1}{2}\|u - y\|_2^2 + \zeta(y)$ denotes the proximal operator of $\zeta$. Analogous to more traditional GNN architectures, (3) contains a $\zeta$-dependent nonlin-

ear activation applied to the output of an affine, graph-dependent filter. With respect to the former, if $\zeta$ is chosen as an indicator function that assigns an infinite penalty to any embedding less than zero, then $\text{prox}_\zeta$ reduces to standard ReLU activations; we will adopt this choice for MuseGNN.

## 2.2 WHY UNFOLDED GNNS?

There are a variety of reasons why it can be advantageous to construct $h$ using unfolded optimization steps as in (3) or related. Of particular note here, UGNN node embeddings inherit exploitable/interpretable characteristics of the lower-level energy (Chen et al., 2021; Zheng et al., 2024), especially if $K$ is sufficiently large such that $Y^{(K)}$ approximates a minimum of $\ell(Y)$. For example, it is well-known that an oversmoothing effect can sometimes cause GNN layers to produce node embeddings that converge towards a non-informative constant (Oono & Suzuki, 2020; Li et al., 2018). This can be understood through the lens of minimizing the the second term of (2) in isolation (Cai & Wang, 2020). The latter can be driven to zero whenever $Y_i^{(K)} = Y_j^{(K)}$ for all $i, j \in \mathcal{V}$, since $\text{tr}[(Y^{(K)})^\top L Y^{(K)}] \equiv \sum_{(i,j)\in\mathcal{E}} \|Y_i^{(K)} - Y_j^{(K)}\|_2^2$. However, it is clear how to design $\ell(Y)$ such that minimizers do not degenerate in this way, e.g., by adding the first term in (2), or related generalizations, it has been previously established that oversmoothing is effectively mitigated, even while spreading information across the graph (Fu et al., 2023; Ma et al., 2021; Pan et al., 2020; Yang et al., 2021; Zhang et al., 2020; Zhu et al., 2021).

UGNNs provide other transparent entry points for customization as well. For example, if an energy function is insensitive to spurious graph edges, then a corresponding GNN architecture constructed via energy function descent is likely be robust against corrupted graphs from adversarial attacks or heterophily (Fu et al., 2023; Yang et al., 2021). More broadly, the flexibility of UGNNs facilitates bespoke modifications of (3) for graph signal denoising (Chen et al., 2021), handling long-range dependencies (Xue et al., 2023), forming connections with the gradient dynamics of physical systems (Di Giovanni et al., 2023) and deep equilibrium models (Gu et al., 2020; Yang et al., 2022), exploiting the robustness of boosting algorithms (Sun et al., 2019), or differentiating the relative importance of features versus network effects in making predictions (Yoo et al., 2023).

## 2.3 SCALABILITY CHALLENGES AND CANDIDATE SOLUTIONS

As benchmarks continue to expand in size (see Table 4), it is no longer feasible to conduct full-graph GNN training using only a single GPU or even single machine, especially so for relatively deep UGNNs. To address such GNN scalability challenges, there are presently two dominant lines of algorithmic workarounds. The first adopts various sampling techniques to extract much smaller computational subgraphs upon which GNN models can be trained in mini-batches. Relevant examples include neighbor sampling (Hamilton et al., 2017; Ying et al., 2018), layer-wise sampling (Chen et al., 2018; Zou et al., 2019), and graph-wise sampling (Chiang et al., 2019; Zeng et al., 2021). For each of these, there exist both online and offline versions, where the former involves randomly sampling new subgraphs during each training epoch, while the latter (Zeng et al., 2021; Gasteiger et al., 2022) is predicated on a fixed set of subgraphs for all epochs. Recent work specific to scaling UGNNs (Li et al., 2022) has incorporated subgraphs based on graph partitioning, i.e., splitting the original full graph into disjoint subgraphs, and using these to approximate gradients involving the full-graph energy. While convergence is established for the forward pass, referred to as an unbiased stochastic proximal-solver (USP), there are no guarantees for the joint forward-backward passes during training, nor empirical comparisons on large graphs; see Appendix B.5 for further discussion of USP and how it differs from MuseGNN.

The second line of work exploits the reuse of historical embeddings, meaning the embeddings of nodes computed and saved during the previous training epoch. In doing so, much of the recursive forward and backward computations required for GNN training, as well as expensive memory access to high-dimensional node features, can be reduced (Chen et al., 2017; Fey et al., 2021; Huang et al., 2024). This technique has recently been applied to training UGNN models via the LazyGNN framework (Xue et al., 2023), as well as somewhat related implicit GNN models (Chen et al., 2022b), although available performance results do not cover large-scale graphs and there are no convergence guarantees. Beyond this (and USP from above), we are unaware of prior work specifically devoted to the scaling and coincident analysis of unfolded GNNs.

## 3 GRAPH-REGULARIZED ENERGY FUNCTIONS INFUSED WITH SAMPLING

Our goal of this section is to introduce a convenient family of energy functions formed by applying graph-based regularization to a set of subgraphs that have been sampled from a given graph of interest. For this purpose, we first present the offline sampling strategy which will undergird our approach, followed by details of energy functional form we construct on top of it for use with MuseGNN. Later, we conclude by analyzing special cases of these sampling-based energies, further elucidating relevant properties and connections with full-graph training.

### 3.1 OFFLINE SAMPLING FOUNDATION

In the present context, offline sampling refers to the case where we sample a fixed set of subgraphs from $\mathcal{G}$ once and store them, which can ultimately be viewed as a form of preprocessing step. More formally, we assume an operator $\Omega : \mathcal{G} \to \{\mathcal{G}_s\}_{s=1}^m$, where $\mathcal{G}_s = \{\mathcal{V}_s, \mathcal{E}_s\}$ is a subgraph of $\mathcal{G}$ containing $n_s = |\mathcal{V}_s|$ nodes, of which we assume $n_s'$ are target training nodes (indexed from 1 to $n_s'$), and $\mathcal{E}_s$ represents the edge set. The corresponding feature and label sets associated with the $s$-th subgraph are denoted $X_s \in \mathbb{R}^{n_s \times d}$ and $T_s \in \mathbb{R}^{n_s' \times d'}$, respectively.

There are several key reasons we employ offline sampling as the foundation for our scalable unfolded GNN architecture development. Firstly, conditioned on the availability of such pre-sampled subgraphs, there is no additional randomness when we use them to replace energy functions dependent on $\mathcal{G}$ and $X$ (e.g., as in (2)) with surrogates dependent on $\{\mathcal{G}_s, X_s\}_{s=1}^m$. Hence we retain a deterministic energy substructure contributing to a more transparent bilevel (upper- and lower-level) optimization process and attendant node-wise embeddings that serve as minimizers. Secondly, offline sampling allows us to conduct formal convergence analysis that is agnostic to the particular sampling operator $\Omega$. In this way, we need not compromise flexibility in choosing a practically-relevant $\Omega$ in order to maintain desirable convergence guarantees. And lastly, offline sampling facilitates an attractive balance between model accuracy and efficiency within the confines of unfolded GNN architectures. As will be shown in Section 6, we can match the accuracy of full-graph training with an epoch time similar to online sampling methods, e.g., neighbor sampling.

### 3.2 ENERGY FUNCTION FORMULATION

To integrate offline sampling into a suitable graph-regularized energy function, we first introduce two sets of auxiliary variables that will serve as more flexible input arguments. Firstly, to accommodate multiple different embeddings for the same node appearing in multiple subgraphs, we define $Y_s \in \mathbb{R}^{n_s \times d}$ for each subgraph index $s = 1, \ldots, m$, as well as $\mathbb{Y} = \{Y_s\}_{s=1}^m \in \mathbb{R}^{(\sum n_s) \times d}$ to describe the concatenated set. Secondly, we require additional latent variables that, as we will later see, facilitate a form of controllable linkage between the multiple embeddings that may exist for a given node (i.e., when a given node appears in multiple subgraphs). For this purpose, we define the latent variables as $M \in \mathbb{R}^{n \times d}$, where each row can be viewed as a shared summary embedding associated with each node in the original/full graph.

We then define our sampling-based extension of (2) for MuseGNN as

$$\ell_{\text{muse}}(\mathbb{Y}, M) := \sum_{s=1}^m \left[ \|Y_s - f(X_s; W)\|_F^2 + \lambda \operatorname{tr}(Y_s^\top L_s Y_s) + \gamma \|Y_s - \mu_s\|_F^2 + \sum_{i=1}^{n_s} \zeta(Y_{s,i}) \right], \quad (4)$$

where $L_s$ is the graph Laplacian associated with $\mathcal{G}_s$, $\gamma \geq 0$ controls the weight of the additional penalty factor, and each $\mu_s \in \mathbb{R}^{n_s \times d}$ is derived from $M$ as follows. Let $I(s, i)$ denote a function that maps the index of the $i$-th node in subgraph $s$ to the corresponding node index in the full graph. For each subgraph $s$, we then define $\mu_s$ such that its $i$-th row satisfies $\mu_{s,i} = M_{I(s,i)}$; per this construction, $\mu_{s,i} = \mu_{s',j}$ if $I(s, i) = I(s', j)$. Consequently, $\{\mu_s\}_{s=1}^m$ and $M$ represent the same overall set of latent embeddings, whereby the former is composed of repeated samples from the latter aligned with each subgraph.

Overall, there are three prominent factors which differentiate (4) from (2):

1. $\ell_{\text{muse}}(\mathbb{Y}, M)$ involves a deterministic summation over a fixed set of sampled subgraphs, where each $Y_s$ is unique while $W$ (and elements of $M$) are shared across subgraphs. As will be discussed further below, this energy actually has the potential to be *more* expressive relative to the full graph form in (2).

2. Unlike (2), the revised energy involves both an expanded set of node-wise embeddings $\mathbb{Y}$ as well as auxiliary summary embeddings $M$. When later forming GNN layers designed to minimize $\ell_{\text{muse}}(\mathbb{Y}, M)$, we must efficiently optimize over *all* of these quantities, which alters the form of the final architecture.

3. The additional $\gamma \|Y_s - \mu_s\|_F^2$ penalty factor acts to enforce dependencies between the embeddings of a given node spread across different subgraphs.

With respect to the latter, it is elucidating to consider minimization of $\ell_{\text{muse}}(\mathbb{Y}, M)$ over $\mathbb{Y}$ with $M$ set to some fixed $M'$. In this case, up to an irrelevant global scaling factor and additive constant, the energy can be equivalently reexpressed as

$$\ell_{\text{muse}}(\mathbb{Y}, M = M') \equiv \sum_{s=1}^{m} \left[ \|Y_s - [f'(X_s; W) + \gamma' \mu_s]\|_F^2 + \lambda' \operatorname{tr}(Y_s^\top L_s Y_s) + \sum_{i=1}^{n_s} \zeta'(Y_{s,i}) \right],$$
(5)

where $f'(X_s; W) := \frac{1}{1+\gamma} f(X_s; W)$, $\gamma' := \frac{\gamma}{1+\gamma}$, $\lambda' := \frac{\lambda}{1+\gamma}$, and $\zeta'(Y_{s,i}) := \frac{1}{1+\gamma} \zeta(Y_{s,i})$. From this expression, we observe that, beyond inconsequential rescalings (which can be trivially neutralized by simply rescaling the original choices for $\{f, \gamma, \lambda, \zeta\}$), the role of $\mu_s$ is to refine the initial base predictor $f(X_s; W)$ with a corrective factor reflecting embedding summaries from other subgraphs sharing the same nodes. Conversely, when we minimize $\ell_{\text{muse}}(\mathbb{Y}, M)$ over $M$ with $\mathbb{Y}$ fixed, we find that the optimal $\mu_s$ for every $s$ is equal to the *mean* embedding for each constituent node across all subgraphs. Hence $M'$ chosen in this way via alternating minimization has a natural interpretation as grounding each $Y_s$ to a shared average representation reflecting the full graph structure. We now consider two limiting special cases of $\ell_{\text{muse}}(\mathbb{Y}, M)$ that provide complementary contextualization.

## 3.3 NOTABLE LIMITING CASES

We first consider setting $\gamma = 0$, in which case the resulting energy completely decouples across each subgraph such that we can optimize each

$$\ell_{\text{muse}}^s(Y_s) := \|Y_s - f(X_s; W)\|_F^2 + \lambda \operatorname{tr}(Y_s^\top L_s Y_s) + \sum_{i=1}^{n_s} \zeta(Y_{s,i}), \quad \forall s$$
(6)

independently, $s = 1, \ldots, m$. Under such conditions, the only cross-subgraph dependency stems from the shared base model weights $W$ which are jointly trained. Hence $\ell_{\text{muse}}^s(Y_s)$ is analogous to a full graph energy as in (2) with $\mathcal{G}$ replaced by $\mathcal{G}_s$. We remark that there exists non-isomorphic graphs such that the induced full-graph energies from (2) have equivalent minima, and yet the corresponding $\{\ell_{\text{muse}}^s(Y_s)\}_{s=1}^m$ from (6) do not; see Appendix C for additional analysis and examples. Hence in this sense the sampling-based energy has the potential to be *more expressive than the original*, and need not be viewed as merely an approximation thereof. In Section 5 we will examine convergence conditions for the full bilevel optimization process over all $Y_s$ and $W$ that follows the $\gamma = 0$ assumption. We have also found that this simplified setting performs well in practice; see Appendix B.2 for ablations.

At the opposite extreme when $\gamma = \infty$, we are effectively enforcing the constraint $Y_s = \mu_s, \forall s$. As such, we can directly optimize all $Y_s$ out of the model leading to the reduced $M$-dependent energy

$$\ell_{\text{muse}}(M) := \sum_{s=1}^{m} \left[ \|\mu_s - f(X_s; W)\|_F^2 + \lambda \operatorname{tr}(\mu_s^\top L_s \mu_s) + \sum_{i=1}^{n_s} \zeta(\mu_{s,i}) \right].$$
(7)

Per the definition of $\{\mu_s\}_{s=1}^m$ and the correspondence with unique node-wise elements of $M$, this scenario has a much closer resemblance to full graph training with the original $\mathcal{G}$. In this regard, as long as a node from the original graph appears in at least one subgraph, then it will have a single, unique embedding in (7) aligned with a row of $M$.

Moreover, we can strengthen the correspondence with full-graph training via the suitable selection of the offline sampling operator $\Omega$. In fact, there exists a simple uniform sampling procedure such that, at least in expectation, the energy from (7) is equivalent to the original full-graph version from (2), with the role of $M$ equating to $Y$. More concretely, we present the following (all proofs are deferred to Appendix E):

**Proposition 3.1.** *Suppose we have $m$ subgraphs $(\mathcal{V}_1, \mathcal{E}_1), \ldots, (\mathcal{V}_m, \mathcal{E}_m)$ constructed independently such that $\forall s = 1, \ldots, m, \forall u, v \in \mathcal{V}, \Pr[v \in \mathcal{V}_s] = \Pr[v \in \mathcal{V}_s \mid u \in \mathcal{V}_s] = p; (i, j) \in \mathcal{E}_s \iff i \in$*

$\mathcal{V}_s, j \in \mathcal{V}_s, (i, j) \in \mathcal{E}$. *Then when* $\gamma = \infty$, *we have* $\mathbb{E}[\ell_{muse}(M)] = mp\,\ell(M)$ *with the* $\lambda$ *in* $\ell(M)$ *rescaled to* $p\lambda$.

Strengthened by this result, we can more directly see that (7) provides an intuitive bridge between full-graph models based on (2) and subsequent subgraph models we intend to build via (4), with the later serving as an unbiased estimator of the former. Even so, we have found that relatively small $\gamma$ values nonetheless work well in practice, possibly (at least in some situations) because the sampling-based energy itself may be more expressive as mentioned earlier.

## 4 FROM SAMPLING-BASED ENERGIES TO THE MUSEGNN FRAMEWORK

Having defined and motivated a family of sampling-based energy functions vis-a-vis (4), we now proceed to derive minimization steps that will serve as GNN model layers that define MuseGNN forward and backward passes as summarized in Algorithm 1. Given that there are two input arguments, namely $\mathbb{Y}$ and $M$, it is natural to adopt an alternating minimization strategy whereby we fix one and optimize over the other, and vice versa.

With this in mind, we first consider optimization over $\mathbb{Y}$ with $M$ fixed. Per the discussion in Section 3.2, when conditioned on a fixed $M$, $\ell_{\mathrm{muse}}(\mathbb{Y}, M)$ decouples over subgraphs. Consequently, optimization can proceed using subgraph-independent proximal gradient descent, leading to the update rule

$$Y_s^{(k)} = \mathrm{prox}_\zeta \left[ Y_s^{(k-1)} - \alpha \left( [(1+\gamma)I + \lambda L_s] Y_s^{(k-1)} - [f(X_s; W) + \gamma \mu_s] \right) \right], \quad \forall s. \tag{8}$$

Here the input argument to the proximal operator is given by a gradient step along (4) w.r.t. $Y_s$. We also remark that execution of (8) over $K$ iterations represents the primary component of a single forward training pass of our proposed MuseGNN framework as depicted on lines 6-8 of Algorithm 1.

We next turn to optimization over $M$ which, as mentioned previously, can be minimized by the mean of the subgraph-specific embeddings for each node. However, directly computing these means is problematic for computational reasons, as for a given node $v$ this would require the infeasible collection of embeddings from all subgraphs containing $v$. Instead, we adopt an online mean estimator with forgetting factor $\rho$. For each node $v$ in the full graph, we maintain a mean embedding $M_v$ and a counter $c_v$. When this node appears in the $s$-th subgraph as node $i$ (where $i$ is the index within the subgraph), we update the mean embedding and counter via

$$M_{I(s,i)} \leftarrow \frac{\rho c_{I(s,i)}}{c_{I(s,i)}+1} M_{I(s,i)} + \frac{(1-\rho)c_{I(s,i)}+1}{c_{I(s,i)}+1} Y_{s,i}^{(K)}, \qquad c_{I(s,i)} \leftarrow c_{I(s,i)} + 1. \tag{9}$$

Also shown on line 9 of Algorithm 1, we update $M$ and $c$ once per forward pass, which serves to refine the effective energy function observed by the core node embedding updates from (8).

For the backward pass, we compute gradients of

$$\mathcal{L}_{\mathrm{muse}}(W, \theta) := \sum_{s=1}^{m} \sum_{i=1}^{n'_s} \mathcal{D}\left( g\left[ Y_s^{(K)}(W)_i; \theta \right], T_{s,i} \right) \tag{10}$$

w.r.t. $W$ and $\theta$ as listed on line 10 of Algorithm 1, where $\mathcal{L}_{\mathrm{muse}}(W, \theta)$ is a sampling-based modification of (1). For $W$ though, we only pass gradients through the calculation of $Y_s^{(K)}$, not the full online $M$ update; however, provided $\rho$ is chosen to be sufficiently large, $M$ will change slowly relative to $Y_s^{(K)}$ such that this added complexity is not necessary for obtaining reasonable convergence.

## 5 CONVERGENCE ANALYSIS OF MUSEGNN

**Global Convergence with $\gamma = 0$.** In the more restrictive setting where $\gamma = 0$, we derive conditions whereby the entire MuseGNN bilevel optimization pipeline converges to a solution that jointly minimizes both lower- and upper-level energy functions in a precise sense to be described shortly. We remark here that establishing convergence is generally harder for bilevel optimization problems relative to more mainstream, single-level alternatives (Colson et al., 2005). To describe our main result, we first require the following definition:

**Definition 5.1.** Assume $f(X; W) = XW$, $\gamma = 0$, $\zeta(y) = 0$, $g(y; \theta) = y$, and that $\mathcal{D}$ is a Lipschitz continuous convex function. Given the above, we then define $\mathcal{L}_{\text{muse}}^{(k)}(W)$ as (10) with $K$ set to $k$. Analogously, we also define $\mathcal{L}_{\text{muse}}^{*}(W)$ as (10) with $Y_s^{(K)}$ replaced by $Y_s^{*} := \arg\min_{Y_s} \ell_{\text{muse}}^s(Y_s)$ for all $s$.

**Theorem 5.2.** *Let $W^*$ be the optimal value of the loss $\mathcal{L}_{muse}^{*}(W)$ per Definition 5.1, while $W^{(t)}$ denotes the value of $W$ after $t$ steps of stochastic gradient descent over $\mathcal{L}_{muse}^{(k)}(W)$ with diminishing step sizes $\eta_t = O(\frac{1}{\sqrt{t}})$. Then provided we choose $\alpha \in \left(0, \min_s \|I + \lambda L_s\|_2^{-1}\right]$ and $Y_s^{(0)} = f(X_s; W)$, for some constant $C$ we have that*

$$\mathbb{E}\left[\mathcal{L}_{muse}^{(k)}(W^{(t)})\right] - \mathcal{L}_{muse}^{*}(W^*) \leq O\left(\frac{1}{\sqrt{t}} + e^{-Ck}\right).$$

---

**Algorithm 1** MuseGNN Training Procedure

---

**Require:** $\{\mathcal{G}_s\}_{s=1}^m$: subgraphs, $\{X_s\}_{s=1}^m$: feat., $K$: # unfolded layers, $E$: # epochs
1: Randomly initialize $W$ and $\theta$;
   Initialize $c \in \mathbb{R}^n$ and $M \in \mathbb{R}^{n \times d}$ to zero
2: **for** $e = 1, 2, \ldots, E$ **do**
3:    **for** $s = 1, 2, \ldots, m$ **do**
4:      $\mu_{s,i} \leftarrow M_{I(s,i)}, i = 1, 2, \ldots, n_s$
5:      $Y_s^{(0)} \leftarrow f(X_s; W)$
6:      **for** $k = 1, 2, \ldots, K$ **do**
7:        Update $Y_s^{(k)}$ using (8)
8:      **end for**
9:      Update $M, c$ via (9)
10:      Update $W, \theta$ via SGD over loss (10)
11:    **end for**
12: **end for**

---

Given that $\mathcal{L}_{\text{muse}}^{*}(W^*)$ is the global minimum of the combined bilevel system, this result guarantees that we can converge arbitrarily close to it with adequate upper- and lower-level iterations, i.e., $t$ and $k$ respectively. Note also that *we have made no assumption on the sampling method*. In fact, as long as offline sampling is used, convergence is guaranteed, although the particular offline sampling approach can potentially impact the convergence rate; see Appendix D.2 for further details.

**Lower-Level Convergence with arbitrary $\gamma$.** We now address the more general scenario where $\gamma \geq 0$ is arbitrary. However, because of the added challenge involved in accounting for alternating minimization over both $\mathbb{Y}$ and $M$, it is only possible to establish conditions whereby the lower-level energy (4) is guaranteed to converge as follows.

**Theorem 5.3.** *Assume $\zeta(y) = 0$. Suppose that we have a series of $\mathbb{Y}^{(k)}$ and $M^{(k)}$, $k = 0, 1, 2, \ldots$ constructed following the updating rules $\mathbb{Y}^{(k)} := \arg\min_Y \ell_{muse}(\mathbb{Y}, M^{(k-1)})$ and $M^{(k)} := \arg\min_M \ell_{muse}(\mathbb{Y}^{(k)}, M)$, with $\mathbb{Y}^{(0)}$ and $M^{(0)}$ initialized arbitrarily. Then*

$$\lim_{k \to \infty} \ell_{muse}(\mathbb{Y}^{(k)}, M^{(k)}) = \inf_{\mathbb{Y}, M} \ell_{muse}(\mathbb{Y}, M). \tag{11}$$

While technically this result does not account for minimization over the upper-level MuseGNN loss, we still empirically find that the entire process outlined by Algorithm 1, including the online mean update, is nonetheless able to converge in practice; see Appendix D.1 example.

## 6 EXPERIMENTS

We now seek to show that MuseGNN serves as a reliable unfolded GNN model that:

1. Preserves competitive accuracy across datasets of widely varying size, especially on the very largest publicly-available graph benchmarks, with a single fixed architecture,
2. Operates with comparable computational complexity relative to common alternatives that are also capable of scaling to the largest graphs, and
3. Empirically converges across different $\gamma$ values that modulate our proposed sampling-based energy, in accordance with theory.

For context though, we note that graph benchmark leaderboards often include top-performing entries based on complex compositions of existing models and training tricks, at times dependent on additional features or external data not included in the original designated dataset. Although these

approaches have merit, our goal herein is not to compete with them, as they typically vary from dataset to dataset. Additionally, they are more frequently applied to smaller graphs using architectures that have yet to be consistently validated across multiple, truly large-scale benchmarks.

Table 2: Node classification accuracy (%) on the test set, except for MAG240M which only has labels for validation set. Bold numbers denote the highest accuracy. For the two largest datasets, MuseGNN is currently the top-performing homogeneous graph model on the relevant OGB-LSC and IGB leaderboards respectively, even while maintaining the attractive interpretability attributes of an unfolded GNN. We omit error bars for baseline results in the two largest datasets because of the high cost to run them all. Additionally, OOM refers to out-of-memory.

| | Small ($|\mathcal{V}|$ <0.5M) | | Medium ($|\mathcal{V}|$ ≈2M) | Large ($|\mathcal{V}|$ ≈100M) | Largest ($|\mathcal{V}|$ >240M) | |
|---|---|---|---|---|---|---|
| **Model** | arxiv | IGB-tiny | products | papers100M | MAG240M | IGB-full |
| GCN (NS) | 69.71±0.25 | 69.77±0.11 | 78.49±0.53 | 65.83±0.36 | 65.24 | 48.59 |
| SAGE (NS) | 70.49±0.20 | 72.42±0.09 | 78.29±0.16 | 66.20±0.13 | 66.79 | 54.95 |
| GAT (NS) | 69.94±0.28 | 69.70±0.09 | 79.45±0.59 | 66.28±0.08 | 67.15 | 55.51 |
| SGFormer (NS) | 70.55±0.24 | **73.37±0.12** | 78.94±0.25 | 66.01±0.37 | 65.29 | N/A |
| MariusGNN | 69.36±0.09 | 73.06±0.06 | 76.95±0.24 | 62.97±0.13 | 63.17 | 54.99 |
| FreshGNN | 71.51±0.03 | 71.49±0.13 | 79.21±0.37 | 66.22±0.07 | 65.53 | 56.50 |
| GCN (GAS) | 71.68±0.3 | 67.86±0.20 | 76.6±0.3 | 54.2±0.7 | OOM | OOM |
| SAGE (GAS) | 71.35±0.4 | 69.35±0.06 | 77.7±0.7 | 57.9±0.4 | OOM | OOM |
| GAT (GAS) | 70.89±0.1 | 69.23±0.17 | 76.9±0.5 | OOM | OOM | OOM |
| UGNN (FG) | **72.74±0.25** | 72.44±0.09 | OOM | OOM | OOM | OOM |
| LazyGNN[1] | 72.30±0.18 | 72.92±0.17 | **81.21±0.21** | 39.80±0.09 | OOM | OOM |
| USP | 71.6±0.2 | N/A | 73.8±0.3 | N/A | N/A | N/A |
| MuseGNN | 72.50±0.19 | **73.42±0.03** | **81.23±0.39** | **66.82±0.02** | **67.67±0.13** | **61.74±0.05** |

**Datasets.** We evaluate the performance of MuseGNN on node classification tasks from the Open Graph Benchmark (OGB) (Hu et al., 2020; 2021) and the Illinois Graph Benchmark (IGB) (Khatua et al., 2023), which are based on homogeneous graphs spanning a wide range of sizes. Table 4 in Appendix A presents the relevant size-related details. Of particular note is IGB-full, currently the largest publicly-available graph benchmark exceeding 1TB in size. We also point out that while MAG240M is originally a heterogenous graph, we follow the common practice of homogenizing it (Hu et al., 2021); similarly for IGB datasets, we adopt the provided homogeneous versions.

**MuseGNN Design.** For *all* experiments, we choose the following fixed MuseGNN settings: Both $f(X; W)$ and $g(Y; \theta)$ are 3-layer MLPs, the number of unfolded layers $K$ is 8 (rationale discussed later below), and the embedding dimension $d$ is 512, and the forgetting factor $\rho$ for the online mean estimation is 0.9. For offline subgraph sampling, we choose a variation on neighbor sampling called ShadowKHop (Zeng et al., 2021), which loosely approximates the conditions of Proposition 3.1. See Appendix A for further details regarding the MuseGNN implementation and hyperparameters.

**Baseline Models.** With the stated objectives of this section in mind, we compare MuseGNN with very recent scalability-focused GNN frameworks MariusGNN (Waleffe et al., 2023), FreshGNN (Huang et al., 2024), and SGFormer (Wu et al., 2024), the latter representing a graph transformer equipped with neighbor sampling[2] and other adaptations explicitly for handling large graphs. Likewise we include GCN (Kipf & Welling, 2017), GAT (Velickovic et al., 2018), and GraphSAGE (Hamilton et al., 2017), in each case testing with both neighbor sampling (NS) (Hamilton et al., 2017) and GNNAutoScale (GAS) (Fey et al., 2021) for scaling to the largest graphs. As a key point of reference, we note that GAT with neighbor sampling is currently the top-performing homogeneous graph model on both the MAG240M and IGB-full leaderboards. Another natural baseline we adopt is the analogous full-graph (FG) UGNN with the same architecture as MuseGNN

---

[1] The accuracy of LazyGNN on arxiv and products is slightly different from what is reported in Xue et al. (2023) for reasons specified in Appendix A.

[2] In Wu et al. (2024), neighbor sampling is only used for larger graphs; however, for consistency across methods and benchmarks in our experiments, we apply sampling to SGFormer for all datasets regardless of size; likewise for MuseGNN and all other sampling-based models.

but without scalable sampling. And finally, we compare with both recent approaches explicitly designed for scaling UGNNs, namely, USP (Li et al., 2022) and LazyGNN (Xue et al., 2023) as discussed in Sections 1 and 2.3. For USP, the core solver is applied to the energy from (2) for the most direct head-to-head comparisons, noting that MuseGNN, LazyGNN, and USP can all potentially be applied to alternative energy functions to improve performance if needed (we use the accuracy reported in the USP paper where available because there is no public code at this time). Further comparisons against other scalable baselines Cluster-GCN (Chiang et al., 2019), GraphFM (Yu et al., 2022), SGC (Wu et al., 2019), and SIGN (Rossi et al., 2020) are deferred to Appendix B.1.

**Accuracy Comparisons.** As shown in Table 2, MuseGNN achieves similar accuracy to a comparable full-graph unfolded GNN model on the small datasets (satisfying our first objective from above), while the latter is incapable of scaling to even the mid-sized `products` benchmark. Meanwhile, MuseGNN is generally better than the other GNN baselines, particularly so on the largest dataset, `IGB-full`. Notably, MuseGNN exceeds the performance of GAT (NS) and FreshGNN, the current SOTA for homogeneous graph models on `MAG240M` and `IGB-full`. Similarly, MuseGNN outperforms all of the additional scalable GNN frameworks presented in Appendix B.1. In terms of prior work scaling UGNNs, public code for USP is presently not available; however, based on reported results from Li et al. (2022) using the same full-graph energy (2), MuseGNN performance is significantly higher; see `arxiv` and `products` results in Table 2. As for LazyGNN, results are competitive on the small- and medium-sized datasets, but deteriorate rapidly on `papers100M` and run OOM on the largest datasets. We note that all methods relying on GAS (including LazyGNN) experience degradation as the data sizes increase. See Appendix A for more USP/LazyGNN details.

Table 3: Training speed (epoch time) in seconds; hardware configurations in Appendix A.

| Model | papers-100M | MAG240M | IGB-full |
|---|---|---|---|
| SAGE (NS) | 102.48 | 795.19 | 17279.98 |
| GAT (NS) | 164.14 | 1111.30 | 19789.79 |
| LazyGNN | 6972.38 | OOM | OOM |
| MuseGNN | 158.85 | 1370.47 | 20413.69 |

**Timing Comparisons.** Turning to model complexity, Table 3 displays the training speed of MuseGNN relative to two common GNN baselines as well as LazyGNN. From these results, we observe that MuseGNN executes with a similar epoch time to the GNNs (satisfying our second objective from above), and much more efficiently relative to LazyGNN. Note that without public code, we are unable to compare timing with USP.

**Convergence Illustration and Ablations.** In Figure 1 we show the empirical convergence of (4) w.r.t. $\mathbb{Y}$ during the forward pass of MuseGNN models on `ogbn-papers100M` for differing values of $\gamma$. In all cases the energy converges within 8 iterations, supporting our choice of $K = 8$ for experiments (and satisfying our third objective). Please see Appendix B for additional ablations involving $\gamma$ and the MuseGNN offline sampler.

## 7 CONCLUSION

In this work, we have proposed MuseGNN, an unfolded GNN model that scales to large datasets by incorporating sampled subgraphs into the design of its lower-level, architecture-inducing energy function. In so doing, MuseGNN readily handles graphs with $\sim 10^8$ or more nodes and high-dimensional node features, exceeding 1TB in total size. Moreover, this is accomplished while maintaining interpretable layers with desirable inductive biases, concomitant convergence guarantees, and competitive (and at times SOTA) accuracy on the largest available benchmarks. The latter significantly exceeds the capabilities of prior efforts to scale UGNNs.

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

# A  EXPERIMENT DETAILS

**Dataset Statistics.**  Table 4 contains the details of the datasets we use for the experiments in Section 6 and the ablation study in Appendix B.2.

Table 4: Dataset details, including node feature dimension (Dim.) and number of classes (# Class).

| Dataset | $|\mathcal{V}|$ | $|\mathcal{E}|$ | Dim. | # Class | Dataset Size |
|---|---|---|---|---|---|
| `ogbn-arxiv` (Hu et al., 2020) | 0.17M | 1.2M | 128 | 40 | 182MB |
| `IGB-tiny` (Khatua et al., 2023) | 0.1M | 0.5M | 1024 | 19 | 400MB |
| `ogbn-products` (Hu et al., 2020) | 2.4M | 123M | 100 | 47 | 1.4GB |
| `IGB-medium` (Khatua et al., 2023) | 10.0M | 120M | 1024 | 19 | 40.8GB |
| `ogbn-papers100M` (Hu et al., 2020) | 111.1M | 1.6B | 128 | 172 | 57GB |
| `MAG240M` (Hu et al., 2021) | 244.2M | 1.7B | 768 | 153 | 377GB |
| `IGB-full` (Khatua et al., 2023) | 269.3M | 4.0B | 1024 | 19 | 1.15TB |

**Model Details.**  For MuseGNN, we choose the base model $f(X;W)$ and output function $g(Y;\theta)$ to be two shallow 3-layer MLPs with standard residual skip connections (although for the somewhat differently-structured `ogbn-products` data we found that skip connections were not necessary). The node-wise constraint $\eta$ in the energy function (4) is set to be non-negativity, making the proximal operator $\text{prox}_\eta$ in the model (8) to be ReLU. The number of unfolded layers, $K$, is selected to be 8, and the hidden dimension $d$ is set to 512. The forgetting factor $\rho$ for the online mean estimation is set to be 0.9. All the models and experiments are implemented in PyTorch (Paszke et al., 2019) using the Deep Graph Library (DGL) (Wang, 2019).

**Offline Samples.**  As supported by Proposition 3.1, subgraphs induced from nodes (where all the edges between sampled nodes are kept) pair naturally with the energy function. Therefore we use node-induced subgraphs for our experiments. For `ogbn-arxiv`, we use the 2-hop full neighbor because it is a relatively small graph. For `IGB-tiny`, `ogbn-papers100M`, and `MAG240M`, we use ShadowKHop with fanout [5,10,15]. For the rest of the datasets, we use ShadowKHop with fanout [10, 15]. Note that ShadowKHop sampler performs node-wise neighbor sampling and returns the subgraph induced by all the sampled nodes.

**Training Hyperparameters.**  In the training process, we set the dropout rate of the MLP layers to be 0.2, and we do not have dropout between the propagation layers. The parameters are optimized by Adam optimizer (Kingma & Ba, 2014) with the weight decay parameter set to 0 and the learning rate being 0.001. For `ogbn-arxiv` and `ogbn-papers100M`, $\alpha = 0.05, \lambda = 20$. For `MAG240M` and the `IGB` series datasets, $\alpha = 0.2, \lambda = 4$. And for `ogbn-products`, $\alpha = \lambda = 1$ with preconditioning on the degree matrix for each unfolded step/layer. The batch size $n'_s$ in the offline samples are all set to 1000. For large-scale datasets, an expensive hyperparameter grid search as commonly used for GNN tuning is not feasible. Hence we merely applied simple heuristics informed from training smaller models to pick hyperparameters for the larger datasets. Also it is worth noticing that $\alpha$ should not in principle affect accuracy if set small enough, since the model will eventually converge. In this regard, $\alpha$ can be set dependent on $\lambda$, since the latter determines the size of $\alpha$ needed for convergence.

**Evaluation Process.**  In the presented results, the training, validation and test set splits all follow the original splits from the datasets. The evaluation process for validation and test datasets uses the same pipeline as the training process does: doing offline sampling first and then use these fixed samples to do the calculation. The only difference is that in the evaluation process, the backward propagation is not performed. While we used the same pipeline for training and testing, MuseGNN is modular and we could optionally use online sampling for the evaluation or even multi-hop full-neighbor loader in the evaluation time.

**Configurations in the Speed Experiments.**  We use a single AWS EC2 p4d.24xlarge instance to run the speed experiments. It comes with dual Intel Xeon Platinum 8275CL CPU (48 cores, 96 threads), 1.1TB main memory and 8 A100 (40GB) GPUs. All the experiments are run on single

GPU. When running MuseGNN, the pre-sampled graph structures are stored in the NVMe SSDs and the input feature is loaded into the main memory (for `IGB-full`, its feature exceeds the capacity of the main memory, so mmap is employed). In the baseline setting, `ogbn-papers100M` and `MAG240M` are paired with neighbor sampling of fanout [5, 10, 15] and `IGB-full` is paired with neighbor sampling of fanout [10, 15]. So the fanout for the online neighbor sampling baselines and MuseGNN are the same so that they can have a fair comparison. In these experiments, the hidden size is set to 256 as is typical.

**Additional Details Regarding USP and LazyGNN Results.** Since the source code for USP is not available, we can only cite the published numbers that are directly applicable in Table 2. For a fair comparison, we choose USP results listed as "IRLS" as the base model in (Li et al., 2022). This model corresponds to minimizing the energy function (2), which is equivalent to what is used for our full-graph unfolded GNN baseline, UGNN (FG), and the original basis for MuseGNN. Note that MuseGNN and USP can both handle alternative energy function forms to potentially improve performance depending on the application.

As for LazyGNN, its training pipeline incorporates METIS partitioning similar to other GAS-based methods. The more partitions used for this, the smaller the subgraphs are (so that they can fit into the GPU memory), and the more iterations it takes for training one epoch. For obtaining `ogbn-arxiv` and `ogbn-products` results, the partitions are only 60 and 150. However, for `ogbn-papers100M`, as this dataset is much larger in size, the number of partitions is set to 40000 so that the subgraphs can fit into GPU memory. But GAS-based LazyGNN also uses historical embeddings and historical gradients, and the staleness of these variables can become very large due to higher number of iterations per epoch needed with so many partitions. Hence this staleness could partially account for the relatively low accuracy on `ogbn-papers100M`. Related discussion of this aspect of GAS-based models can be found in Huang et al. (2024). And finally, for obtaining the accuracy results of LazyGNN, we ran the exact open-sourced code[1] with the hyper-parameters provided by the authors for each overlapping dataset they considered (i.e., `arxiv` and `products`), and found that it is actually *maximum accuracy* values (over 5 trials) that matches what is reported in their paper. However, for consistency within our experiments across all methods, we instead report the LazyGNN *mean accuracy* values with error bars in Table 2.

# B    ADDITIONAL EXPERIMENTS AND DISCUSSION

## B.1    ACCURACY COMPARISONS WITH ADDITIONAL BASELINES

For a more general comparison beyond Table 2 in the main paper, this section introduces additional baselines, and in particular, some recent models explicitly designed for scalability (although none of these models are UGNNs). Specifically, we include SGC (Wu et al., 2019), SIGN (Rossi et al., 2020), Cluster-GCN (Chiang et al., 2019), and GraphFM (Yu et al., 2022). Note that GraphFM is built on top of GAS (Fey et al., 2021), while the others (i.e., SGC, SIGN, ClusterGCN) represent traditional GNN architectures that are naturally amenable to larger graphs. Results using these additional baselines, as presented in Table 5, further solidify the competitive advantage of MuseGNN. Additionally, all baseline results are from OGB leaderboards or published papers (Huang et al., 2024; Zhu & Koniusz, 2020). In this regard, we do not report performance on IGB datasets as, being quite new, most prior work does not contain such results and it is extremely expensive to run ourselves.

## B.2    FURTHER DETAILS REGARDING THE ROLE OF $\gamma$ IN MUSEGNN

The incorporation of $\gamma > 0$ serves two purposes within our MuseGNN framework and its supporting analysis. First, by varying $\gamma$ we are able to build a conceptual bridge between the decoupled $\gamma = 0$ base scenario, and full-graph training as $\gamma \to \infty$, provided the sampled subgraphs adhere to the conditions of Proposition 3.1. In this way, the generality of $\gamma > 0$ has value in terms of elucidating connections between modeling regimes, independently of empirical performance.

---

[1] https://github.com/RXPHD/Lazy_GNN/

Table 5: Node classification accuracy (%) results using additional non-UGNN baselines. Reported numbers are from the test set, except for `MAG240M` which only has labels for the validation set.

| Model | arxiv | products | papers100M | MAG240M |
|---|---|---|---|---|
| SGC | 68.78 | 68.87 | 63.29 | 65.82 |
| SIGN | 71.95 | 80.52 | 65.68 | 66.64 |
| Cluster-GCN | 68.11 | 78.97 | 53.35 | OOM |
| GraphFM | 71.53 | 70.76 | 48.03 | OOM |
| MuseGNN | **72.50** | **81.23** | **66.82** | **67.67** |

That being said, the second role is more pragmatic, as $\gamma > 0$ can indeed improve predictive accuracy. As shown in the ablation in Table 6 (where all hyperparameters except $\gamma$ remain fixed), $\gamma = 0$ already achieves good performance; however, increasing $\gamma$ leads to further improvements. This is likely because larger $\gamma$ enables the model to learn similar embeddings for the same node in different subgraphs, which may be more robust to the sampling process.

Table 6: Ablation study of the penalty factor $\gamma$. Results shown represent the accuracy (%) on the test set. Bold numbers denote the best performing method.

| | $\gamma = 0$ | $\gamma = 0.1$ | $\gamma = 0.5$ | $\gamma = 1$ | $\gamma = 2$ | $\gamma = 3$ |
|---|---|---|---|---|---|---|
| papers100M | 66.29 | 66.31 | 66.53 | 66.45 | 66.64 | **66.82** |
| ogbn-arxiv | 72.51 | **72.67** | 72.47 | 72.21 | 72.15 | 72.08 |
| IGB-tiny | 72.66 | 72.78 | **73.42** | 73.02 | 72.70 | 72.69 |
| IGB-medium | 75.18 | 75.80 | **75.83** | N/A | N/A | N/A |
| ogbn-products | 80.42 | 80.92 | **81.23** | N/A | N/A | N/A |

That being said, while we can improve accuracy with $\gamma > 0$, this generality comes with an additional memory cost for storing the required mean vectors. However, this memory complexity for the mean vectors is only $O(nd)$, which is preferable to the memory complexity $O(ndK)$ of GAS (Fey et al., 2021), which depends on the number of layers $K$. Importantly, this more modest $O(nd)$ storage is only an upper bound for the MuseGNN memory complexity, as the $\gamma = 0$ case is already very effective even without requiring additional storage. As an additional point of reference, the GAS-related approach from Xue et al. (2023) also operates with only single-layer $O(nd)$ of additional storage; however, unlike MuseGNN, this storage is mandatory for large graphs (not amenable to full-graph training) and therefore serves as a *lower* bound for memory complexity. Even so, the approach from Xue et al. (2023) is valuable and complementary.

### B.3 Sampling Ablation on the Baseline Models

We choose neighbor sampling for the baselines because it is most commonly used, but since ShadowKHop is paired with MuseGNN, we use Table 7 to show that the improvement is not coming from the change in the sampling method but the usage of energy-based scalable unfolded model. In Table 7, the baseline models are trained on `ogbn-papers100M` with the same offline ShadowKHop samples used in the training of MuseGNN, but they suffer from a decrease in the accuracy compared with the online neighbor sampling counterparts, so the change in sampling method cannot account for the enhancement in the accuracy.

Table 7: Ablation study of the accuracy improvement compared with baselines. Shown results are the accuracy (%) on the test set of `ogbn-papers100M`.

| | Baseline Model | | |
|---|---|---|---|
| Sampling Method | GCN | GAT | SAGE |
| ShadowKHop (offline) | 64.89 | 63.84 | 65.57 |
| Neighbor Sampling (online) | 65.83 | 66.28 | 66.2 |

### B.4 Alternative Sampling Method for MuseGNN

We also paired MuseGNN with online neighbor sampling, the most popular choice for sampling, on `ogbn-papers100M` to account for our design choice of sampling methods coupled with MuseGNN. Theoretically, the convergence guarantee no longer holds true for online sampling. Additionally, neighbor sampling removes terms regarding edge information from the energy function, so we expect it to be worse than the offline ShadowKHop sampling that we use. This is also supported by Proposition 3.1 because neighbor sampling does not provide subgraphs induced from the nodes with all the edges. Empirically, with online neighbor sampling, the test accuracy is 66.11%; it is indeed lower than the 66.82% from our offline ShadowKHop results.

### B.5 Differences between MuseGNN and USP

Despite both being methods for scaling UGNN training, the USP approach (Li et al., 2022) and MuseGNN differ in multiple important respects. First, USP is based on an unbiased stochastic proximal solver and partitioned subgraphs to approximate the gradient steps used for minimizing an *original* full-graph energy as in (2). Quite differently, MuseGNN is actually predicated on defining a new subgraph-based energy function given by (4) that is distinct from its full-graph counterpart. See Section C below for discussion of the potentially greater expressiveness of (4) relative to (2).

Secondly, USP convergence has only been established for the forward pass (i.e., lower-level energy function minimization), with no guarantees provided for full convergence in conjunction with the backward pass (i.e., minimizing the upper-level training loss w.r.t. model parameters). In contrast, we establish MuseGNN convergence under certain conditions across the full bilevel optimization process, simultaneously covering forward and backward passes.

And lastly, there does not as of yet exist any published demonstration that USP is empirically effective on large graph datasets, nor public code available for implementing such a demonstration. In fact, `ogbn-products` is the largest benchmark adopted in Li et al. (2022), which is viewed as medium-sized relative to the other datasets we consider. Note that full-graph GNNs can be effectively trained on `ogbn-products` using a single GPU, so this dataset does not generally require intensive scalability measures. This is unlike MuseGNN, which we have validated on the largest publicly-available benchmarks. We also note that Li et al. (2022) presents a compelling acceleration method to reduce training iterations; however, the per-iteration computational complexity of this step is comparable to full-graph training and therefore is not suitable for the largest graphs.

## C MuseGNN Expressiveness Relative to Full-Graph Unfolded GNNs

In this section we consider the expressiveness of MuseGNN versus full graph unfolded GNN models. As suggested in Section 3, there exists non-isomorphic graphs such that the induced full-graph energies from (2) have effectively equivalent minima, and yet the corresponding $\{\ell_{\text{muse}}^s(Y_s)\}_{s=1}^m$ from (6), as a special case of the MuseGNN energy (4) with $\gamma = 0$, do not. This would imply that the subgraph-based energy of MuseGNN can actually be *more* expressive in some sense. We first formalize this notion and then subsequently provide illustrative examples.

### C.1 Formal Analysis

The following proposition demonstrates that for two non-isomorphic graphs that are considered as isomorphic by the Weisfeiler-Lehman (WL) test (Shervashidze et al., 2011), minimizing the full-graph energy (2) will yield equivalent embeddings, even when initialized in such a way that the actual descent iterations need not follow the WL criteria which underpin the test itself.

**Proposition C.1.** *Suppose two attributed graphs $\mathcal{G}_1 = (\mathcal{V}_1, \mathcal{E}_1, X_1)$ and $\mathcal{G}_2 = (\mathcal{V}_2, \mathcal{E}_2, X_2)$, although possibly non-isomorphic, cannot be distinguished as non-isomorphic by the WL test. Furthermore, w.l.o.g. assume the node indices of $\mathcal{G}_1$ and $\mathcal{G}_2$ are aligned such that each node with the same index has the same final hash value produced by the WL test, i.e., the multi-set of their neighborhoods are identical (if this were not the case, it can be trivially achieved via permutation of node indices, an operation that does not impact the energy (2)). Now consider optimizing (2) using either*

*graph with a convex $\zeta$. Beginning from **arbitrary** initial embeddings $Y_1^{(0)}$ and $Y_2^{(0)}$ respectively, not necessarily equal to one another nor adhering to WL test stipulations, the unfolded GNN descent iterations from (3) will converge to minimizing solutions $Y_1^*$ and $Y_2^*$ satisfying $Y_1^* = Y_2^*$.*

*Proof.* Since $\mathcal{G}_1$ and $\mathcal{G}_2$ are considered isomorphic by WL test, we have $X_1 = X_2$. Let $X_1 = X_2 = X$. The energy functions for $\mathcal{G}_1$ and $\mathcal{G}_2$ are then $\ell_1(Y) = \|Y - f(X; W)\|_F^2 + \lambda \operatorname{tr}(Y^\top L_1 Y) + \sum_{i=1}^n \zeta(Y_i)$ and $\ell_2(Y) = \|Y - f(X; W)\|_F^2 + \lambda \operatorname{tr}(Y^\top L_2 Y) + \sum_{i=1}^n \zeta(Y_i)$ respectively, where $L_1$ and $L_2$ denote the graph Laplacians associated with $\mathcal{G}_1$ and $\mathcal{G}_2$.

We first prove that *within one graph*, the nodes with the same final WL hash value will have the same embedding. Since here the energy function is strongly convex, the initialization can be arbitrary and the forward pass will finally reach the same global minimum. Now suppose we initialize the embeddings as $f(X; W)$. In this case, the nodes with same WL hash values have the same initial embeddings. Now consider each proximal gradient descent step on the energy function, i.e., the forward propagation layer of the unfolded GNN. As the neighborhood multi-sets of the nodes with same WL hash are exactly the same, for these nodes in every step, the aggregation and update functions take in exactly the same inputs. Therefore in every step these nodes have the same embeddings, thus at the final minimum, the embeddings are still the same. And since any initialization leads to the same minimum, this will be true regardless of the descent steps that led there.

Now we prove the proposition regarding two graphs. Considering the first order optimality condition of the energy $\ell_1(Y)$, we have $-(Y_1^* - f(X; W) + \lambda L Y_1^*) \in \partial \zeta(Y_1^*)$, where $\partial \zeta(Y_1^*)$ denotes the subdifferential of $\zeta$ evaluated at $Y = Y_1^*$. With the assumption on the alignment of node indices, we want to show that $Y_1^* = Y_2^*$, which is equivalent to $Y_1^*$ satisfying the first order condition of $\ell_2(Y)$. Therefore, it is sufficient to show when $-(Y_1^* - f(X; W) + \lambda L_2 Y_1^*) \in \partial \zeta(Y_1^*) \Rightarrow -(Y_1^* - f(X; W) + \lambda L_1 Y_1^*) \in \partial \zeta(Y_1^*)$. Since multiplying the Laplacian matrix and the embeddings is essentially doing one step of neighbor aggregation, we have $L_1 Y_1^* = L_2 Y_1^*$. This is because the 1-hop neighborhood of each node in $\mathcal{G}_1$ and $\mathcal{G}_2$ have the same hash value combinations as guaranteed by the WL test, and each node with same hash value has the same embedding in $Y_1^*$ as we proved above. Therefore $-(Y_1^* - f(X; W) + \lambda L_1 Y_1^*) = -(Y_1^* - f(X; W) + \lambda L_2 Y_1^*)$, so they both exist in $\partial \zeta(Y_1^*)$. Thus, $Y_1^*$ is also the minimum for $\ell_2(Y)$, so $Y_1^* = Y_2^*$. $\qquad\square$

Interestingly though, something equivalent to Proposition C.1 need *not* hold for the MuseGNN energy based on sampled subgraphs. To see this, we rely on analysis from (Frasca et al., 2022), which reveals that the expressive power of GNNs trained on subgraphs can actually surpass the expressive power of the WL test. In our setting, it can be shown that this translates into more expressive minimizers of the MuseGNN energy (4), particularly when $\gamma = 0$. To intuitively convey this phenomena, we next lean on illustrative examples whereby minimizers of (4) can distinguish two graphs that were not originally distinguishable by the WL test or minimizers of (2).

## C.2 ILLUSTRATIVE EXAMPLES

To begin, we consider pairs of non-isomorphic graphs that have been initially assigned identical input features across all nodes in both graphs. For simplicity of presenting useful illustrative examples, we also base our assumed energy functions on symmetric normalized graph Laplacians $\tilde{L} = D^{-\frac{1}{2}} L D^{-\frac{1}{2}} = I - D^{-\frac{1}{2}} A D^{-\frac{1}{2}}$. Note that Proposition C.1 still holds equally well under this revision.

Starting with uniform initial node features, the WL test will ultimately identify the two pairs of non-isomorphic graphs in Figure 2a as isomorphic. And with the WL test result, we can match each node in the left graph with a corresponding node in the right graph if they have the same final hash value in the WL test. A bijection can be established for such corresponding nodes because these two graphs are considered isomorphic by the WL test.

If we use the left-right pair of graphs in Figure 2a and uniform initial features to form the energy (2), subsequent optimization using (3) will produce the final embedding minimizers as indicated by the node colors in Figure 2a. Here the same color indicates the same embedding vector of the minimizer. As the corresponding nodes from the left-right pair have the same colors, the minimizers of the corresponding full-graph energies are equivalent. Thus the left-right pair cannot be distinguished. Such cases resonate with the implications of Proposition C.1.

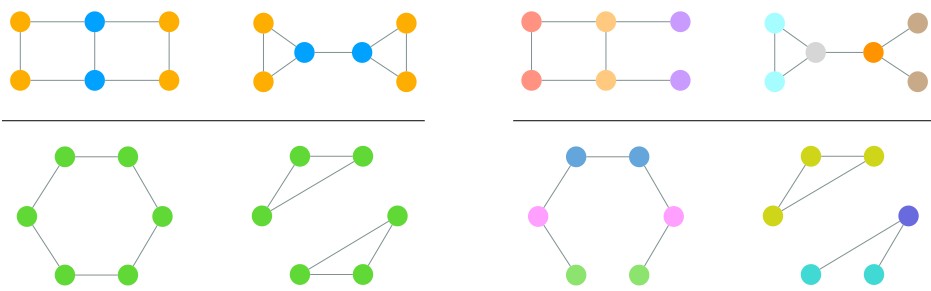

(a) Same colors indicate that these nodes have the same embedding vector upon minimizing the full-graph energy from (2) using the corresponding graphs. The non-isomorphic graphs within each left-right pair *cannot* be differentiated.

(b) Based on subgraphs of the original graphs on the left, minimizers of a revised energy akin to (6) and by extension (4) can in fact now differentiate each non-isomorphic graph, as noted by the distinct color patterns within each left-right pair.

Figure 2: Building on analysis from Frasca et al. (2022), it is possible to achieve increased expressiveness via energy functions based on sampled subgraphs (as incorporated by MuseGNN).

However, if we sample subgraphs from the full graphs, we are now able to distinguish such pairs. By taking the subgraphs of the left-right pair in Figure 2a, we obtain the graph structures in Figure 2b. But now, even with the uniform node features, the final embeddings minimizing the graph energy (6) will yield different distributions for the left-right pair, thus differentiating the two graphs. The final embeddings are illustrated as colors in Figure 2b. As such, the MuseGNN energy function (4) based on sampled subgraphs is in this sense more expressive than the full-graph energy alternative from (2), as minimizers of the former can differentiate some non-isomorphic graph pairs that are not distinguishable by the minimizers of the latter.

## D  ADDITIONAL CONVERGENCE DETAILS

### D.1  EMPERICAL CONVERGENCE RESULTS

Here we show the empirical results of the previous convergence analysis. Though we do not have the full bilevel convergence results for the $\gamma > 0$ case like Theorem 5.2, Figure 3 shows that in real-world dataset, the bilevel optimization system will still converge with the imprecise estimation of optimal embeddings $Y_s^{(K)} \approx Y_s^*$ and the online mean estimation of $M$.

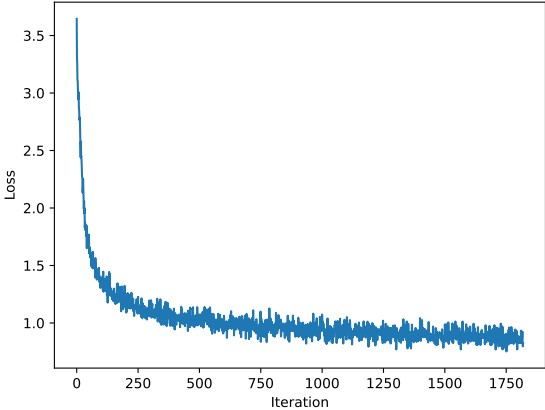

Figure 3: Convergence of the upper-level loss on `ogbn-arxiv` dataset for 20 epochs with penalty factor $\gamma = 1$.

### D.2 A NOTE ON SAMPLING METHODS IMPACTING CONVERGENCE RATES

As stated in the main paper, offline sampling allows us to conduct formal convergence analysis that is agnostic to the particular sampling operator. So our convergence guarantee is valid for all sampling methods. However, different sampling methods can still actually have an impact on the convergence *rate* of the bilevel optimization. This is because different sampling methods will produce subgraphs with different graph Laplacian matrices, and the latter may have differing condition numbers $\frac{\sigma}{\tau}$ as mentioned later in Lemma E.4. In this regard, a larger such condition number will generally yield a slower convergence rate. Further details are elaborated in the proof of Theorem 5.2 in Appendix E.2.

## E PROOFS

### E.1 CONNECTION WITH FULL GRAPH TRAINING

**Proposition 3.1.** *Suppose we have $m$ subgraphs $(\mathcal{V}_1, \mathcal{E}_1), \ldots, (\mathcal{V}_m, \mathcal{E}_m)$ constructed independently such that $\forall s = 1, \ldots, m, \forall u, v \in \mathcal{V}, \Pr[v \in \mathcal{V}_s] = \Pr[v \in \mathcal{V}_s \mid u \in \mathcal{V}_s] = p; (i, j) \in \mathcal{E}_s \iff i \in \mathcal{V}_s, j \in \mathcal{V}_s, (i, j) \in \mathcal{E}$. Then when $\gamma = \infty$, we have $\mathbb{E}[\ell_{muse}(M)] = mp\,\ell(M)$ with the $\lambda$ in $\ell(M)$ rescaled to $p\lambda$.*

*Proof.* The probability that an edge is sampled is $\Pr[(i, j) \in \mathcal{E}_s \mid (i, j) \in \mathcal{E}] = p^2$. $\forall v \in \mathcal{V}$, if $v \in \mathcal{V}_s, v \in \mathcal{V}_t$, then $Y_{s,v} = Y_{t,v} = M_v$. We have

$$\ell_{\text{muse}}(M) = \sum_{s=1}^{m} \left[ \sum_{v \in \mathcal{V}_s} \left( \|M_v - f(X; W)_v\|_2^2 + \zeta(M_v) \right) + \frac{\lambda}{2} \sum_{(i,j) \in \mathcal{E}_s} \|M_i - M_j\|_2^2 \right]$$

$$\mathbb{E}\left[\ell_{\text{muse}}(M)\right] = \sum_{s=1}^{m} \sum_{v \in \mathcal{V}_s} \left( \|M_v - f(X; W)_v\|_2^2 + \zeta(M_v) \right) \cdot \Pr[v \in \mathcal{V}_s]$$

$$+ \frac{\lambda}{2} \sum_{s=1}^{m} \sum_{(i,j) \in \mathcal{E}_s} \|M_i - M_j\|_2^2 \cdot \Pr[(i, j) \in \mathcal{E}_s]$$

$$= mp \sum_{v \in \mathcal{V}} \left( \|M_v - f(X; W)_v\|_2^2 + \zeta(M_v) \right) + mp^2 \frac{\lambda}{2} \sum_{(i,j) \in \mathcal{E}} \|M_i - M_j\|_2^2$$

$$= mp \left[ \|M - f(X; W)\|_F^2 + p\lambda \operatorname{tr}\left(M^\top L M\right) + \sum_{i=1}^{n} \zeta(M_i) \right]$$

$$= mp\,\ell(M)$$

$\square$

### E.2 FULL CONVERGENCE ANALYSIS

According to Definition 5.1, $Y_s^*(W) = (I + \lambda L_s)^{-1} f(X_s; W) := P_s^* f(X_s; W)$ is the optimal embedding for the subgraph energy $\ell_{\text{muse}}^s(Y_s)$. Additionally, we initialize $Y_s^{(0)}$ as $f(X_s; W)$, so $Y_s^{(k)}$ can be written as $P_s^{(k)} f(X; W)$ where $P_s^{(k)}$ is also a matrix. We have $\alpha \leq \|I + \lambda L_s\|_2^{-1}$, so $P_s^{(k)}$ will converge to $P_s^*$ as $k$ grows. The output of $\mathcal{D}$ over matrix input is the sum of $\mathcal{D}$ over each row vectors, as typical discriminator function like squared error or cross-entropy do.

Plugging the $Y_s^{(k)}(W)$ and $Y_s^*(W)$ into the loss function, we have

$$\mathcal{L}_{\text{muse}}^{(k)}(W) = \sum_{s=1}^{m} \mathcal{D}\left(Y_s^{(k)}(W)_{1:n_s'}, T_s\right) = \mathcal{D}\left(\mathbb{P}^{(k)} \mathbb{X} W, \mathbb{T}\right) \tag{12}$$

where $1 : n'_s$ means the first $n'_s$ rows, $\mathbb{T} := (T_1, T_2, \ldots, T_m)^\top$, $\mathbb{P}^{(k)} := \text{diag}\left\{\left(P_s^{(k)}\right)_{1:n'_s}\right\}_{s=1}^m$.

Similarly, for $\ell_W^*(W)$, we change all the $Y_s^{(k)}$ and $P_s^{(k)}$ to $Y_s^*$ and $P_s^*$. That is,

$$\mathbb{P}^* := \text{diag}\left\{(P_s^*)_{1:n'_s}\right\}_{s=1}^m, \quad \mathcal{L}_{\text{muse}}^*(W) = \mathcal{D}\left(\mathbb{P}^*\mathbb{X}W, \mathbb{T}\right)$$

**Stochastic Gradient Descent** The updating rule by gradient descent for the parameter is

$$W^{(t+1)} = W^{(t)} - \eta \nabla_{W^{(t)}} \mathcal{L}_{\text{muse}}^{(k)}(W^{(t)})$$

$$= W^{(t)} - \eta \sum_{s=1}^m \frac{\partial \mathcal{D}\left(Y_s^{(k)}(W^{(t)}), T_s\right)}{\partial W^{(t)}}$$

In reality, we use stochastic gradient descent to minimize $\mathcal{L}_{\text{muse}}^{(k)}(W)$, and the updating rule becomes

$$W^{(t+1)} = W^{(t)} - \eta \frac{\partial \mathcal{D}\left(T_s, g\left(Y_s^{(k)}(W^{(t)})\right)\right)}{\partial W^{(t)}} \tag{13}$$

Here $s$ is picked at random from $\{1, 2, \ldots, m\}$, so the gradient is an unbiased estimator of the true gradient.

**Lemma E.1.** *As defined in* (12)*, $\mathcal{L}_{muse}^{(k)}(W)$ is a convex function of $W$. Furthermore, there exist the global optimal $W^{(k*)}$ such that $\mathcal{L}_{muse}^{(k)}(W^{(k*)}) \leq \mathcal{L}_{muse}^{(k)}(W)$ for all $W$.*

*Proof.* $\mathcal{L}_{\text{muse}}^{(k)}(W)$ is a composition of convex function $\mathcal{D}$ with an affine function, so the convexity is retained. By the convexity, the global optimal $W^{(k*)}$ exists. $\square$

**Lemma E.2.** *When the loss function $\mathcal{L}_{muse}^{(k)}$ is defined as equation* (12)*, and the parameter $W$ is updated by* (13) *with diminishing step sizes $\eta_t = O(\frac{1}{\sqrt{t}})$, $\mathbb{E}\left[\mathcal{L}_{muse}^{(k)}(W^{(t)})\right] - \mathcal{L}_{muse}^{(k)}(W^{(k*)}) = O(1/\sqrt{t})$.*

*Proof.* Since the loss function $\mathcal{L}_{\text{muse}}^{(k)}$ is convex by Lemma E.1, and the step size is diminishing in $O(\frac{1}{\sqrt{t}})$, by (Nemirovski et al., 2009), the convergence rate of the difference in expected function value and optimal function value is $O(\frac{1}{\sqrt{t}})$. $\square$

**Lemma E.3.** *The subgraph energy function* (6) *with $\zeta(y) = 0$ (as defined in Definition 5.1) is $\sigma_s$-smooth and $\tau_s$-strongly convex with respect to $Y_s$, with $\sigma_s = \sigma_{\max}(I + \lambda L_s)$ and $\tau_s = \sigma_{\min}(I + \lambda L_s)$, where $\sigma_{\max}$ and $\sigma_{\min}$ are the maximum and minimum singular value of the matrix.*

*Proof.* The proof is simple by computing the Hessian of the energy function. Additionally, since the graph Laplacian matrix $L_s$ is positive-semidefinite, we have $\sigma_s \geq \tau_s > 0$. $\square$

**Lemma E.4.** *Let $\sigma, \tau = \arg\max_{\sigma_s, \tau_s} \frac{\sigma_s}{\tau_s}, s = 1, 2, \cdots, m$, where $\frac{\sigma}{\tau}$ gives the worst condition number of all subgraph energy functions. In the descent iterations from* (8) *that minimizes* (6) *with step size $\alpha = \frac{1}{\sigma}$, we can establish the bound on the propagation matrix $\|\mathbb{P}^{(k)} - \mathbb{P}^*\| \leq m \exp(-\frac{\tau}{2\sigma}k) \sum_{s=1}^m \|P_s^{(0)} - P_s^*\|$*

*Proof.* By Bubeck et al. (2015)[Theorem 3.10], we have

$$\left\|P_s^{(k)} f(X; W) - P_s^* f(X; W)\right\|^2 \leq e^{-\frac{\tau_s}{\sigma_s}k} \left\|P_s^{(0)} f(X; W) - P_s^* f(X; W)\right\|^2$$

Since this bound holds true for any $f(X; W)$, by choosing $f(X; W) = I$ to be the identity matrix, we have

$$\left\|P_s^{(k)} - P_s^*\right\| \leq e^{-\frac{\tau_s}{2\sigma_s}k} \left\|P_s^{(0)} - P_s^*\right\| \leq e^{-\frac{\tau}{2\sigma}k} \left\|P_s^{(0)} - P_s^*\right\|$$

Adding up all the $m$ subgraphs, we have

$$\left\| \mathbb{P}^{(k)} - \mathbb{P}^* \right\| \leq m e^{-\frac{\tau}{2\sigma}k} \sum_{s=1}^{m} \left\| P_s^{(0)} - P_s^* \right\|$$

$\square$

**Theorem 5.2.** *Let $W^*$ be the optimal value of the loss $\mathcal{L}_{muse}^*(W)$ per Definition 5.1, while $W^{(t)}$ denotes the value of $W$ after $t$ steps of stochastic gradient descent over $\mathcal{L}_{muse}^{(k)}(W)$ with diminishing step sizes $\eta_t = O(\frac{1}{\sqrt{t}})$. Then provided we choose $\alpha \in \left(0, \min_s \|I + \lambda L_s\|_2^{-1}\right]$ and $Y_s^{(0)} = f(X_s; W)$, for some constant $C$ we have that*

$$\mathbb{E}\left[\mathcal{L}_{muse}^{(k)}(W^{(t)})\right] - \mathcal{L}_{muse}^*(W^*) \leq O\left(\frac{1}{\sqrt{t}} + e^{-Ck}\right).$$

*Proof.* In Definition 5.1, we assume $\mathcal{D}$ to be Lipschitz continuous for the embedding variable, so for any input $\mathbb{Y}$ and $\mathbb{Y}'$, we have $|\mathcal{D}(\mathbb{Y}, \mathbb{T}) - \mathcal{D}(\mathbb{Y}', \mathbb{T})| \leq L_\mathcal{D} \|\mathbb{Y} - \mathbb{Y}'\|$.

$$\mathcal{L}_{\text{muse}}^{(k)}\left(W^{(k*)}\right) - \mathcal{L}_{\text{muse}}^*\left(W^*\right)$$
$$\leq \mathcal{L}_{\text{muse}}^{(k)}\left(W^*\right) - \mathcal{L}_{\text{muse}}^*\left(W^*\right)$$
$$= \mathcal{D}\left(\mathbb{P}^{(k)}\mathbb{X}W^*, \mathbb{T}\right) - \mathcal{D}\left(\mathbb{P}^*\mathbb{X}W^*, \mathbb{T}\right)$$
$$\leq L_\mathcal{D} \left\| \mathbb{P}^{(k)}\mathbb{X}W^* - \mathbb{P}^*\mathbb{X}W^* \right\|$$
$$\leq L_\mathcal{D} \left\| \mathbb{P}^{(k)} - \mathbb{P}^* \right\| \|\mathbb{X}W^*\|$$
$$\leq O\left(e^{-\frac{\tau}{2\sigma}k}\right)$$

Therefore,

$$\mathbb{E}\left[\mathcal{L}_{\text{muse}}^{(k)}(W^{(t)})\right] - \mathcal{L}_{\text{muse}}^*(W^*)$$
$$= \mathbb{E}\left[\mathcal{L}_{\text{muse}}^{(k)}(W^{(t)})\right] - \mathcal{L}_{\text{muse}}^{(k)}(W^{(k*)}) + \mathcal{L}_{\text{muse}}^{(k)}(W^{(k*)}) - \mathcal{L}_{\text{muse}}^*(W^*)$$
$$\leq O(\frac{1}{\sqrt{t}}) + O(e^{-\frac{\tau}{2\sigma}k})$$
$$\leq O(\frac{1}{\sqrt{t}} + e^{-ck}),$$

where $c = \frac{\tau}{2\sigma}$. $\square$

### E.3 ALTERNATING MINIMIZATION

The main reference for alternating minimization is Csiszár & Tusnády (1984). In this paper, they proved that the alternating minimization method will converge to the optimal value when a five-point property or both a three-point property and a four-point property holds true. We will show that the global energy function and the corresponding updating rule satisfy the three-point property and the four-point property. Thus, the alternating minimization method will converge to the optimal value.

For simplicity, we define $r_{s,i}$ as $r_{s,i} = \sum_{s'=1}^{m} \sum_{j=1}^{n'_s} \mathbb{I}\{I(s,i) = I(s',j)\}$, where $\mathbb{I}\{\cdot\}$ is the indicator function. Therefore, $r_{s,i}$ means the node in the full graph with index $I(s,i)$ appears $r_{s,i}$ times in all subgraphs. With some abuse of the notation, we let $r_{I(s,i)} = r_{s,i}$.

In the energy function (4), we still want to find a set of $\{Y_s\}_{s=1}^{m}$ and $\{\mu_s\}_{s=1}^{m}$ to minimize the global energy function. We can use the alternating minimization method to solve this problem. In each step, we minimize $\{Y_s\}_{s=1}^{m}$ first and then minimize $\{\mu_s\}_{s=1}^{m}$. We want to show that when the

parameter $W$ is fixed, by alternatively minimizing $\{Y_s\}_{s=1}^m$ and $\{\mu_s\}_{s=1}^m$, the energy will converge to the optimal value.

We can easily get the updating rule for $\{Y_s\}_{s=1}^m$ and $\{\mu_s\}_{s=1}^m$ by taking the derivative of the energy function. Note that we always first update $\{Y_s\}_{s=1}^m$ and then update $\{\mu_s\}_{s=1}^m$. For $\{Y_s\}$, we have

$$Y_s^{(k)} = [(1+\gamma)I + \lambda L_s]^{-1}[f(X_s; W) + \gamma\mu_s^{(k-1)}] \tag{14}$$

For $\{\mu_s\}_{s=1}^m$, we have the $i$-th row of $\mu_s^{(k)}$ is

$$\mu_{s,i}^{(k)} = \frac{1}{r_{s,i}} \sum_{s'=1}^m \sum_{j=1}^{n_s'} Y_{s',j}^{(k)} \cdot \mathbb{I}\{I(s,i) = I(s',j)\} \tag{15}$$

Namely, the updated $\mu^{(k)}$ is the average of the embeddings of the same node in different subgraphs.

Note that here $\{Y_s\}$ with $\{\mu_s\}$ and $\mathbb{Y}$ with $M$ are used simultaneously. We will use $\{Y_s\}$ and $\{\mu_s\}$ when we want to emphasize the subgraphs and use $\mathbb{Y}$ and $M$ when we want to emphasize them as the input of the global energy function. But in essence they represent the same variables and can be constructed from each other.

The three-point property and four-point property call for a non-negative valued helper function $\delta(Y, Y')$ such that $\delta(Y, Y) = 0$. We define the helper function as $\delta(Y, Y') = (1+\gamma)\|Y - Y'\|_F^2$. We can easily verify that $\delta(Y, Y) = 0$ and $\delta(Y, Y') \geq 0$.

**Lemma E.5** (Three-point property). *Suppose that we have a series of $\mathbb{Y}^{(k)}$ and $M^{(k)}$, $k = 0, 1, 2, \cdots$ constructed following the updating rule (14) and (15) and that they are initialized arbitrarily. For any $\mathbb{Y}$ and $k$, $\ell_{muse}(\mathbb{Y}, \mu^{(k)}) - \ell_{muse}(\mathbb{Y}^{(k+1)}, \mu^{(k)}) \geq \delta(\mathbb{Y}, \mathbb{Y}^{(k+1)})$ holds true.*

*Proof.* We only need to show $\ell_{\text{muse}}^s(Y_s, \mu_s^{(k)}) - \ell_{\text{muse}}^s(Y_s^{(k+1)}, \mu_s^{(k)}) \geq \delta(Y_s, Y_s^{(k+1)})$. By iterating $s$ from 1 to $m$ and adding the $m$ inequalities together, we can get the desired result.

$$\ell_{\text{muse}}(Y_s, \mu_s^{(k)}) - \ell_{\text{muse}}(Y_s^{(k+1)}, \mu_s^{(k)}) - \delta(Y_s, Y_s^{(k+1)})$$

$$= \|Y_s - f(X_s; W)\|_F^2 + \lambda\operatorname{tr}\left(Y_s^\top L_s Y_s\right) + \gamma\left\|Y_s - \mu_s^{(k)}\right\|_F^2 - \left\|Y_s^{(k+1)} - f(X_s; W)\right\|_F^2$$

$$\quad - \lambda\operatorname{tr}\left(Y_s^{(k+1)\top} L_s Y_s^{(k+1)}\right) - \gamma\left\|Y_s^{(k+1)} - \mu_s^{(k)}\right\|_F^2 - (1+\gamma)\left\|Y_s - Y_s^{(k+1)}\right\|_F^2$$

$$= \left\|Y_s - Y_s^{(k+1)} + Y_s^{(k+1)} - f(X_s; W)\right\|_F^2 + \lambda\operatorname{tr}\left(Y_s^\top L_s Y_s\right)$$

$$\quad + \gamma\left\|Y_s - Y_s^{(k+1)} - Y_s^{(k+1)} - \mu_s^{(k)}\right\|_F^2 - \left\|Y_s^{(k+1)} - f(X_s; W)\right\|_T^2$$

$$\quad - \lambda\operatorname{tr}\left(Y_s^{(k+1)\top} L_s Y_s^{(k+1)}\right) - \gamma\left\|Y_s^{(k+1)} - \mu_s^{(k)}\right\|_F^2 - (1+\gamma)\left\|Y_s - Y_s^{(k+1)}\right\|_F^2$$

$$= 2\left\langle Y_s - Y_s^{(k+1)}, Y_s^{(k+1)} - f(X_s; W)\right\rangle + 2\gamma\left\langle Y_s - Y_s^{(k+1)}, Y_s^{(k+1)} - \mu_s^{(k)}\right\rangle$$

$$\quad + \lambda\operatorname{tr}\left(Y_s^\top L_s Y_s\right) - \lambda\operatorname{tr}\left(Y_s^{(k+1)\top} L_s Y_s^{(k+1)}\right)$$

$$= 2\left\langle Y_s - Y_s^{(k+1)}, (1+\gamma)Y_s^{(k+1)} - \left(f(X_s; W) + \gamma\mu_s^{(k)}\right)\right\rangle + \lambda\left\langle Y_s - Y_s^{(k+1)}, L_s\left(Y_s + Y_s^{(k+1)}\right)\right\rangle$$

$$= \left\langle Y_s - Y_s^{(k+1)}, 2\left(\lambda L_s + (1+\gamma)I\right)Y_s^{(k+1)} - 2\left(f(X_s; W) + \gamma\mu_s^{(k)}\right) + \lambda L_s\left(Y_s - Y_s^{(k+1)}\right)\right\rangle$$

$$= \left\langle Y_s - Y_s^{(k+1)}, \lambda L_s\left(Y_s - Y_s^{(k+1)}\right)\right\rangle \geq 0$$

$\square$

**Lemma E.6** (four-point property). *Suppose that we have a series of $\mathbb{Y}^{(k)}$ and $M^{(k)}$, $k = 0, 1, 2, \cdots$ constructed following the updating rule (14) and (15) and that they are initialized arbitrarily. For any $\mathbb{Y}$, $\mu$ and $k$, $\delta(\mathbb{Y}, \mathbb{Y}^{(k)}) \geq \ell_{muse}(\mathbb{Y}, \mu^{(k)}) - \ell_{muse}(\mathbb{Y}, \mu)$ holds true.*

*Proof.* Expanding all the functions we can get our target is equivalent to

$$\sum_{s=1}^{m}(1+\gamma)\left\|Y_s - Y_s^{(k)}\right\|^2 \geq \gamma\sum_{s=1}^{m}\left(\left\|Y_s - \mu_s^{(k)}\right\|_F^2 - \|Y_s - \mu_s\|_F^2\right)$$

We can actually show a stronger result, that is

$$\sum_{s=1}^{m}\left\|Y_s - Y_s^{(k)}\right\|_F^2 \geq \sum_{s=1}^{m}\left(\left\|Y_s - \mu_s^{(k)}\right\|_F^2 - \|Y_s - \mu_s\|_F^2\right)$$

By the updating rule for $\mu_s$, we have

$$\sum_{s=1}^{m}\left\langle \mu_s - \mu_s^{(k)}, \mu_s^{(k)} - Y_s^{(k)}\right\rangle = \sum_{s=1}^{m}\sum_{i=1}^{n_s}\left\langle \mu_{s,i} - \mu_{s,i}^{(k)}, \mu_{s,i}^{(k)} - Y_{s,i}^{(k)}\right\rangle$$

$$= \sum_{v=1}^{n}\sum_{s=1}^{m}\sum_{i=1}^{n_s}\mathbb{I}\{I(s,i)=v\}\left\langle M_v - M_v^{(k)}, M_v^{(k)} - Y_{s,i}^{(k)}\right\rangle \tag{16}$$

$$= \sum_{v=1}^{n}r_v\left\langle M_v - M_v^{(k)}, M_v^{(k)} - \frac{1}{r_v}\sum_{s=1}^{m}\sum_{i=1}^{n_s}\mathbb{I}\{I(s,i)=v\}Y_{s,i}^{(k)}\right\rangle = 0$$

Therefore,

$$\sum_{s=1}^{m}\left\|Y_s - Y_s^{(k)}\right\|_F^2 = \sum_{s=1}^{m}\left(\|Y_s - \mu_s\|_F^2 + \left\|\mu_s - Y_s^{(k)}\right\|_F^2 + 2\left\langle Y_s - \mu_s, \mu_s - Y_s^{(k)}\right\rangle\right)$$

$$= \sum_{s=1}^{m}\left(\|Y_s - \mu_s\|_F^2 + \left\|\mu_s - \mu_s^{(k)}\right\|_F^2 + \left\|\mu_s^{(k)} - Y_s^{(k)}\right\|_F^2 + 2\left\langle Y_s - \mu_s, \mu_s - Y_s^{(k)}\right\rangle\right)$$

$$+ \sum_{s=1}^{m}2\left\langle \mu_s - \mu_s^{(k)}, \mu_s^{(k)} - Y_s^{(k)}\right\rangle$$

$$\overset{(16)}{=} \sum_{s=1}^{m}\left(2\|Y_s - \mu_s\|_F^2 + \left\|Y_s - \mu_s^{(k)}\right\|_F^2 + \left\|\mu_s^{(k)} - Y_s^{(k)}\right\|_F^2\right)$$

$$+ 2\sum_{s=1}^{m}\left(\left\langle Y_s - \mu_s, \mu_s - Y_s^{(k)}\right\rangle + \left\langle \mu_s - Y_s, Y_s - \mu_s^{(k)}\right\rangle\right)$$

$$= \sum_{s=1}^{m}\left(2\|Y_s - \mu_s\|_F^2 + \left\|Y_s - \mu_s^{(k)}\right\|_F^2 + \left\|\mu_s^{(k)} - Y_s^{(k)}\right\|_F^2\right)$$

$$+ \sum_{s=1}^{m}\left(-2\|Y_s - \mu_s\|_F^2 + 2\left\langle Y_s - \mu_s, \mu_s^{(k)} - Y_s^{(k)}\right\rangle\right)$$

$$= \sum_{s=1}^{m}\left(\left\|Y_s - \mu_s^{(k)}\right\|_F^2 + \left\|\mu_s^{(k)} - Y_s^{(k)}\right\|_F^2 + 2\left\langle Y_s - \mu_s, \mu_s^{(k)} - Y_s^{(k)}\right\rangle\right)$$

$$= \sum_{s=1}^{m}\left(\left\|Y_s - \mu_s^{(k)}\right\|_F^2 + \left\|\mu_s^{(k)} - Y_s^{(k)} + Y_s' - \mu_s\right\|_F^2 - \|Y_s - \mu_s\|_F^2\right)$$

$$\geq \sum_{s=1}^{m}\left(\left\|Y_s - \mu_s^{(k)}\right\|_F^2 - \|Y_s - \mu_s\|_F^2\right)$$

□

**Theorem 5.3.** *Assume $\zeta(y) = 0$. Suppose we have a series of $\mathbb{Y}^{(k)}$ and $M^{(k)}$, $k = 0, 1, 2, \cdots$ constructed following the updating rule (14) and (15), with $\mathbb{Y}^{(0)}$ and $M^{(0)}$ initialized arbitrarily. Then*

$$\lim_{k\to\infty}\ell_{muse}(\mathbb{Y}^{(k)}, M^{(k)}) = \inf_{\mathbb{Y},M}\ell_{muse}(\mathbb{Y}, M),$$

*Proof.* By Lemma E.5 and Lemma E.6 where both the three-point and four-point property hold true, the theorem is obtained according to Csiszár & Tusnády (1984). ∎

## F LIMITATIONS

While MuseGNN was primarily designed for scaling the most common UGNNs on homogeneous graphs, there do exist prior UGNNs models based on energy functions sensitive to heterogeneous graph structure. The basic idea is to update the trace term in Equation (4) to include additional trainable weight matrices that serve to align embeddings of nodes of different types and relationships. HALO (Ahn et al., 2022) is one such example of this. But there is nothing to prevent us from extending the core techniques and analysis that undergird MuseGNN to scale such heterogeneous cases.

## G IMPACT STATEMENT

While we do not envision that there are any noteworthy risks introduced by our work, it is of course always possible that a large-scale model like MuseGNN could be deployed, either intentionally or unintentionally, in such a way as to cause societal harm. For example, a graph neural network designed to screen fraudulent credit applications could be biased against certain minority groups.

