# OpenReview forum: "MuseGNN: Forming Scalable, Convergent GNN Layers that Minimize a Sampling-Based Energy"
_ICLR.cc/2025/Conference — ICLR 2025 Poster_

### Official Review · Reviewer_ddwX · 2024-10-28

**Soundness:** 3
**Presentation:** 3
**Contribution:** 3
**Rating:** 6
**Confidence:** 3

**Summary:**

This paper introduces a scalable graph neural network (GNN) architecture that minimizes an energy function through a sampling-based approach, allowing for interpretable node embeddings optimized for tasks like node classification.

**Strengths:**

1. The paper discusses a critical issue of scalability when adding energy regularization to GCN, where the cost of energy calculation becomes enormous for large-scale graphs and previous methods fail to work on large graphs.

2. The paper proposes a new method, MuseGNN, to enable energy-regularized GNN to efficiently handle large-scale graphs. The paper also comprehensively discusses the optimization procedure and convergence analysis of their method.


3. The paper validates the performance and efficiency of MuseGNN by experiments on multiple-datasets, including large graph datasets where previous methods need huge computational cost.

**Weaknesses:**

1. The annotations and formulation in 2.1 may be confusing for new readers. The authors could adopt similar annotations in Descent Steps of a Relation-Aware Energy Produce Heterogeneous Graph Neural Networks by Ahn et al, which distinguishes the node embedding by basic model and the embedding by energy optimization by different annotations y and y*, and specify the procedure of UGCN to make the problem formulation clearer to a broader audience.
2. The authors could compare the time and memory cost of MuseGNN with more baselines that also uses energy regularization without sampling to strengthen the superiority of MuseGNN.
3. In the introduction, the authors could provide more specified discussions about the advantages of applying energy regularization to GNN to show the importance of this work.

**Questions:**

1. The ablation study of hyperparameter gamma shows that MuseGNN preserves high accuracy when gamma=0. In this case, the embedding of the same node in different subgraphs are not aligned. Can the node embeddings in subgraphs capture the global information of the original graph? Are there any theoretical insights about why MuseGNN still shows comparative performance even when gamma = 0?


2. What is the sampling strategy when sampling subgraphs from the large original graph?

---

> ### Author Response · Authors · 2024-11-18
> **Response to Reviewer ddwX**
>
> Thanks for pointing out positive aspects of our paper along with other constructive feedback.  We address these points in turn as follows.
>
> **Comment:**
> *\[Weakness 1\] The annotations and formulation in 2.1 may be confusing for new readers. The authors could adopt similar annotations in Descent Steps of a Relation-Aware Energy Produce Heterogeneous Graph Neural Networks by Ahn et al, which distinguishes the node embedding by basic model and the embedding by energy optimization by different annotations y and y\*, and specify the procedure of UGCN to make the problem formulation clearer to a broader audience.*
>
> **Response:**
> We chose the current notation to emphasize the correspondence between UGNN descent iterations and generic GNN layers (with $K$ representing both) up front, but we agree that there are other notational possibilities.  Thanks for the suggestion, we can easily make changes if it leads to better readability.
>
>
> **Comment:**
> *\[Weakness 2\] The authors could compare the time and memory cost of MuseGNN with more baselines that also uses energy regularization without sampling to strengthen the superiority of MuseGNN.*
>
> **Response:**
> Good question.  Although there are many variants, full-graph UGNNs mostly follow the basic form of Equation (2), with subsequent branching from there that generally adds further complexity.  So the UGNN OOM we observe in Table 2 is basically inherited by all the full-graph UGNNs we are aware of, i.e., more complex models (such as Yang et al., 2021) require comparable or more memory.  For example, consider the ogbn-papers100M graph benchmark: existing full-graph UGNNs will take at least 1.7TB of GPU memory for the forward propagation during the training, while MuseGNN takes only about 978MB.
>
>
> **Comment:**
> *\[Weakness 3\] In the introduction, the authors could provide more specified discussions about the advantages of applying energy regularization to GNN to show the importance of this work.*
>
> **Response:**
> Section 2.2 provides detailed motivation for UGNN models; however, per the reviewers suggestion we could re-organize some of this material within the Section 1 introduction, space permitting.
>
>
> **Comment:**
> *\[From Question 1\] The ablation study of hyperparameter gamma shows that MuseGNN preserves high accuracy when gamma=0. In this case, the embedding of the same node in different subgraphs are not aligned. Can the node embeddings in subgraphs capture the global information of the original graph?*
>
> **Response:**
> The node embeddings within each subgraph can still capture some global information even when $\gamma=0$, noting that weight parameters are shared across different subgraphs.  And in fact, per our analysis in Section 3.3 and Appendix C, there are also specific advantages in terms of expressiveness that occur by breaking into separate subgraphs.
>
>
> **Comment:**
> *\[From Question 1\] Are there any theoretical insights about why MuseGNN still shows comparative performance even when gamma = 0?*
>
> **Response:**
> Yes, theoretical support for the $\gamma = 0$ case comes from two sources.  First, via Theorem 5.2 we establish convergence guarantees for the full bilevel optimization process when $\gamma = 0$.  And secondly, from Section 3.3 and Proposition C.1 in Appendix C, we demonstrate that the resulting decoupled energy that occurs with $\gamma = 0$ is more expressive than the full-graph alternative (which occurs when $\gamma$ becomes arbitrarily large).
>
>
> **Comment:**
> *\[Question 2\] What is the sampling strategy when sampling subgraphs from the large original graph?*
>
> **Response:**
> For details of the sampling strategy, please refer to Lines 457-458 in the main paper as well as Lines 781-787 in Appendix A.

---

> ### Comment · Area_Chair_XrRX · 2024-11-26
>
> Please check if the authors' response addresses your concerns.

---

### Official Review · Reviewer_pWkD · 2024-11-02

**Soundness:** 3
**Presentation:** 2
**Contribution:** 3
**Rating:** 5
**Confidence:** 4

**Summary:**

This paper developed one subgraph sampling strategy induced from the energy regularization process over the GNN learning process. The author also discussed the convergence and expressive power of their proposed model under some conditions (i.e., $\gamma$). The newly proposed model is advanced with a shared weight matrix over all subgraphs and by the consideration of replicated nodes in multiple subgraphs. Empirical studies show promising results via large-scale graph datasets.

**Strengths:**

1. The paper resolved the scalability problem on the GNNs induced by their energy optimization form. Empirical studies show the proposed model outperforms many baselines via different graph sizes up to 1TB.

2. The paper provided theoretical analysis on the convergence of the proposed model under different values of $\gamma$, which controls the degree of dependency between subgraphs.

**Weaknesses:**

1. Some important analysis and empirical study results are missing and shall be presented via the main text, please see the question part.

2. The organization of the paper needs to be carefully adjusted.

**Questions:**

At the current stage, my questions are as follows:

1. The organization of the paper needs to be improved. For example, figure 1 on page 2 is referred to in both the introduction section and the experiment section (ablation study), which is rarely to be observed in other studies. Also, the ablation result for $\gamma$, located in Table 6, in Appendix B2, can be put into the current main page, which only contains 9.5 pages.

2. In addition to this, I think the ablation on $\gamma$ is very important as this shows how the degree of dependence between subgraphs affects the final training. Therefore, it is recommended that a more detailed analysis of this aspect be input.

3. Another important empirical verification is the performance of the model via different sampling strategies since this directly affects the power of the term $\|Y_s - \mu_s\|$. I found the related content in the Appendix, but it is recommended that the author put some convincing evidence to the main content.

4. Furthermore, the energy regularizer $\lambda \mathrm{Tr}(\mathbf Y_s \mathbf L_s \mathbf Y_x )$ controls the degree of smoothing via each subgraph, and the quantity of $\lambda$ is shared over all the subgraph. However, in real practice, this may not be ideal as subgraphs shall have different degrees of smoothing.

5. The expressive power of the GNN is usually discussed under the graph pooling tasks (e.g., graph level classification), it would be better and interesting to show some pooling results from the proposed model.

Minor modification.

1. In row 183, table 4 is missing, and it is allocated in the appendix.

---

> ### Author Response · Authors · 2024-11-18
> **Response to Reviewer pWkD (Part I)**
>
> Thanks for acknowledging the soundness and the quality of our contribution.  The primary criticism more relates to the paper organization and ablations, particularly a stated preference for certain Appendix details being moved to the main body. These suggestions are easy to fix, and we address them point-by-point below.
>
> **Comment:**
> *\[Weaknesses and Question 1\]
> Some important analysis and empirical study results are missing and shall be presented via the main text, please see the question part.
> The organization of the paper needs to be improved. For example, figure 1 on page 2 is referred to in both the introduction section and the experiment section (ablation study), which is rarely to be observed in other studies. Also, the ablation result for $\gamma$, located in Table 6, in Appendix B2, can be put into the current main page, which only contains 9.5 pages.*
>
> **Response:**
> Figure 1 was included upfront in the introduction simply because it highlights the convergence of our MuseGNN model, a key differentiating characteristic of UGNNs.  Still, this is a stylistic choice (akin to teaser figures commonly used in CVPR papers), and we are happy to consider alternative placements.
>
> As for additional ablation material, etc., we can easily move material from the appendices to the main paper.  Note that while the ICLR guidelines allow for 10 pages submissions, they also state a preference for 9 pages where possible.  Hence we sought to compress our original draft as much as we could, arriving at 9.5 pages.  Of course we can push out to 10 pages as needed to accommodate reviewer suggestions.
>
>
> **Comment:**
> *\[Question 2\] I think the ablation on $\gamma$ is very important as this shows how the degree of dependence between subgraphs affects the final training. Therefore, it is recommended that a more detailed analysis of this aspect be input.*
>
> **Response:**
> Table 6 in Appendix B.2 contains a detailed ablation over four of the datasets.  We did not include the largest ones because an ablation on these is extremely expensive.  Increasing $\gamma$ beyond the boldface max values shown on the table will make the performance starts to slightly decrease, so we just did not include them.  Ogbn-arxiv was not shown in the table, but we found that the performance is best with $\gamma=0.1$ at 72.50%.  For $\gamma=0$, it is 72.29%, $\gamma=0.5$ is 72.38%, $\gamma=1$ is 72.27%, and $\gamma=2$ is 72.15%.  Does the reviewer have something else particular in mind for the ablation?  We can use the remaining discussion period to run it.
>
>
> **Comment:**
> *\[Question 3\] Another important empirical verification is the performance of the model via different sampling strategies since this directly affects the power of the term $\|Y_s-\mu_s\|$. I found the related content in the Appendix, but it is recommended that the author put some convincing evidence to the main content.*
>
> **Response:**
> Per our comments above, we are happy to move relevant ablations from the appendices to the main text, space permitting.  The original rationale for deferring such material to the appendices was merely for compressing closer to 9 pages as much as possible, as suggested by the ICLR guidelines.
>
>
>
> **Comment:**
> *\[Question 4\] Furthermore, the energy regularizer $\lambda tr(Y_s^\top L_sY_s)$ controls the degree of smoothing via each subgraph, and the quantity of $\lambda$ is shared over all the subgraph. However, in real practice, this may not be ideal as subgraphs shall have different degrees of smoothing.*
>
> **Response:**
> We chose a single, fixed $\lambda$ in part to preserve symmetries with existing full-graph UGNN models, and in part to maintain a simple, easily-tunable pipeline (one which we have shown works well in practice).  That being said, it is useful to consider more flexible alternatives as the reviewer suggests.  In this regard, one notable option is to include a separate $\lambda$ for each subgraph, and then train them all (just like other model parameters $W$ and $\theta$) over the upper-level loss from Equation (10).  Actually, we have tried this approach but found that it does not consistently improve performance.  For example, see results on obgn-arxiv and IGB-tiny below:
>
> | Dataset  | Original | Separate $\lambda$ |
> | -------- | -------- | -------- |
> |ogbn-arxiv| 72.50%   | 71.73%   |
> | IGB-tiny | 73.42%   | 72.34%   |
>
> Still, this is a great suggestion, and could potentially be advantageous in future, more complex use cases.  We can certainly add a mention of this possibility to the paper.

---

> > ### Author Response · Authors · 2024-11-18
> > **Response to Reviewer pWkD (Part II)**
> >
> > **Comment:**
> > *\[Question 5\] The expressive power of the GNN is usually discussed under the graph pooling tasks (e.g., graph level classification), it would be better and interesting to show some pooling results from the proposed model.*
> >
> > **Response:**
> > In principle UGNNs can be applied to graph-level classification tasks.  However, as our focus is exclusively on scaling UGNNs to huge graphs, graph classification is currently out of our scope.  This is because the graphs that comprise typical graph classification benchmarks are extremely tiny (often with only 10s or 100s of nodes); see for example (Yanardag et al., ''Deep graph kernels,'' KDD 2015), and (Dwivedi et al., ''Long range graph benchmark,'' NeurIPS 2022). Note that the average number of nodes in the *largest* graphs from the Dwivedi et al. benchmarks is only 479, while we target graphs with hundreds of millions of nodes.

---

> > > ### Author Response · Authors · 2024-11-25
> > > **Follow-up to Reviewer pWkD**
> > >
> > > The reviewer's primary concerns related to paper organization (e.g., moving more appendix material to the main paper) and ablation details (e.g., varying $\gamma$).  As our rebuttal has addressed these issues,  we are checking now whether or not the reviewer had any follow-up questions/comments before the discussion period ends?

---

> > > > ### Comment · Reviewer_pWkD · 2024-11-26
> > > > **Response to Authors**
> > > >
> > > > Thank you for your time for the rebuttal. Your response resolved most of my questions. My major concern left is Question 4, which is related to the subgraph smoothing problem. I can see that using a fixed $\lambda$ for all the subgraphs could work well in your examples. However, this could also depend on the size of your subgraph and its related node features, as the effect of energy regularization is on every subgraph.
> > > >
> > > > Based on your analysis, I can only find that a larger number of subgraphs (i.e., $m$) leads to a slower convergence. Could the author provide more discussions on the impact of $m$ on your fixed $\lambda$? Please let me know if you have provided the number of subgraphs in your experiment.

---

> > > > > ### Author Response · Authors · 2024-11-27
> > > > > **Follow-up Response to Reviewer pWkD**
> > > > >
> > > > > Thanks for continuing to engage with our paper and thoughtfully considering our rebuttal.  We address follow-up comments point-by-point below, and are happy to discuss further if additional clarification is needed.
> > > > >
> > > > > **Comment:**
> > > > > *I can see that using a fixed $\lambda$ for all the subgraphs could work well in your examples. However, this could also depend on the size of your subgraph and its related node features, as the effect of energy regularization is on every subgraph.*
> > > > >
> > > > > **Response:**
> > > > > Note that we randomly sample subgraphs from the full graph, and each of these random samples serves as an estimator for the full graph (we are not partitioning the original graph into disjoint subgraphs).  In fact, using the sampling method described in Proposition 3.1 will produce subgraphs whose Laplacians are unbiased estimators of the full graph Laplacian up to a constant (the expectation here is over sampling, and holds even with $m=1$).  Hence these subgraphs are capable of loosely mimicking the energy regularization of the full graph, and so a single, shared $\lambda$ is tenable.  These conclusions are also validated by the new experiments mentioned in our rebuttal where we compare against training a separate $\lambda$ for each subgraph.
> > > > >
> > > > > **Comment:**
> > > > > *I can only find that a larger number of subgraphs (i.e., $m$) leads to a slower convergence. Could the author provide more discussions on the impact of $m$ on your fixed $\lambda$? Please let me know if you have provided the number of subgraphs in your experiment.*
> > > > >
> > > > > **Response:**
> > > > > Just to clarify, a larger $m$ need not impact the convergence rate.  As stated in Theorem 5.2, the convergence rate of the bilevel MuseGNN optimization process is determined by the training steps (i.e., $t$) of the upper-level loss and the number of propagation layers from the lower-level energy (i.e., $k$).
> > > > >
> > > > > As for the impact of $m$ on $\lambda$, there is no appreciable relationship we are aware of.  This is because $\lambda$ determines the trade-off between regularizing node and network effects, and at least in expectation, this trade-off is shared across each sampled subgraph (also, observe from Equation (4) that for each included subgraph, a corresponding node regularization term and Laplacian smoothing term are added in tandem, preserving the balance). Therefore we do not find it necessary to set $\lambda$ as a function of $m$.
> > > > >
> > > > >
> > > > > Still, although $m$ is not a key factor influencing $\lambda$ or convergence, we can nonetheless provide further details regarding how $m$ is chosen. Basically, we adopt a training batch size of 1000 for all experiments (see Appendix A), and so the number of subgraphs for a dataset is simply the number of training nodes divided by 1000.  For example, this amounts to 60 subgraphs for IGB-tiny, 91 for ogbn-arxiv, and 1208 for ogbn-papers100M.

---

> > > > > > ### Author Response · Authors · 2024-12-02
> > > > > >
> > > > > > Given that the timeframe in which reviewers can respond will expire in less than a day, was there any further clarification requested by the reviewer?  If so please let us know, as ICLR grants authors another full day to write a final reply.

---

> ### Comment · Area_Chair_XrRX · 2024-11-26
>
> Please check if the authors' response addresses your concerns.

---

### Official Review · Reviewer_RrLb · 2024-11-04

**Soundness:** 3
**Presentation:** 3
**Contribution:** 3
**Rating:** 6
**Confidence:** 3

**Summary:**

This paper introduces MuseGNN, a novel Graph Neural Network (GNN) architecture designed to address the challenges of scaling and convergence in GNNs. The core idea behind MuseGNN is to iteratively minimize a sampling-based energy function during the forward pass, which allows the node embeddings to serve dual purposes: as predictive features for downstream tasks and as minimizers of the energy function. The authors present a scalable GNN that is able to deal with large-scale node classification benchmarks, including datasets exceeding 1TB in size.

**Strengths:**

- The paper provides an analysis of the convergence properties of the proposed energy function and the iterative reduction process.
- MuseGNN is designed to handle very large graphs, as evidenced by its performance on the node classification benchmark exceeding 1TB in size.
- The experimental results show that MuseGNN achieves competitive accuracy compared to state-of-the-art GNN.

**Weaknesses:**

- The paper focuses on comparing MuseGNN with a few specific GNN architectures and frameworks. A more comprehensive comparison with a wider range of state-of-the-art GNNs. In particular, I want to see the comparison to SGC [1] and its variants in terms of efficiency and accuracy.
- The training speed of the proposed method seems to be slower than GAT as shown in Table 1, while GAT is a well-known slow GNN.

[1] Simplifying Graph Convolutional Networks

**Questions:**

- How is the method compared to SGC and its variants in terms of efficiency and accuracy?
- How does the method perform on heterophilic graphs? The energy-related loss seems to highly rely on the homophilic assumption.

---

> ### Author Response · Authors · 2024-11-18
> **Response to Reviewer RrLb**
>
> Thanks for the positive feedback and acknowledging the soundness of our work.
>
> **Comment:**
> *\[Weakness 1 and Question 1\] The paper focuses on comparing MuseGNN with a few specific GNN architectures and frameworks. A more comprehensive comparison with a wider range of state-of-the-art GNNs. In particular, I want to see the comparison to SGC [1] and its variants in terms of efficiency and accuracy.*
>
> **Response:**
> It is well-known that simplified architectures like SGC can be quite efficient in practice, since $A^r X$ only needs to be computed once (where $r$ is a small integer), followed by standard logistic regression or MLP training for node classification.  However, this simplicity often comes with a significant drop in accuracy depending on problem complexity.  In Appendix B.1 we have compared MuseGNN with SGC and additional GNN architectures, and find that SGC accuracy falls well behind MuseGNN.
>
>
> Although it is not generally feasible for any standard GNN approach (e.g., GCN, SAGE, GAT, etc.) to outperform SGC in terms of efficiency, nor is it the goal of MuseGNN to do so either, we provide a comparison here on ogbn-papers100m for reference.  As SGC can be viewed as a full-graph method, we have tested time-to-convergence, finding that SGC takes 2286s, SAGE with neighbor sampling takes 3075s, and MuseGNN takes 5213s.   As expected SGC is the fastest, but again, our primary goal is to scale UGNNs, not reduce computational cost relative to the most lightweight alternatives.
>
>
> **Comment:**
> *\[Weakness 2\] The training speed of the proposed method seems to be slower than GAT as shown in Table 1, while GAT is a well-known slow GNN.*
>
> **Response:**
> Indeed GAT is not generally the fastest GNN architecture; however, for the largest graph datasets it achieves very high accuracy (second only to MuseGNN in some cases) so we view it as a relevant baseline for comparison.  We emphasize though that in Table 3 we also compare against SAGE, a widely-used architecture known for its efficiency and scalability when paired with neighbor sampling.  And although MuseGNN is not quite as efficient as SAGE, it is at least comparable.  This is sufficient to achieve our primary objective of scaling UGNN models with reasonable computational complexity while maintaining high accuracy.  And as a final point of reference, arguably the strongest existing scalable UGNN alternative is LazyGNN, which falls well behind (see Tables 2 and 3).
>
>
> **Comment:**
> *\[Question 2\] How does the method perform on heterophilic graphs? The energy-related loss seems to highly rely on the homophilic assumption.*
>
> **Response:**
> Presently, there are no large-scale heterophilic graph benchmarks that we are aware of, so we have not specifically targeted this question.  However, in the past it has been shown that UGNNs are able to effectively handle heterophily (on smaller graphs) through modifications of the lower-level energy; specifically, the distance metric used to penalize deviations between neighboring node embeddings is modified (Yang et al., 2021; Ahn et al., 2022).  MuseGNN could likewise adopt similar changes to accommodate large-scale heterophilic graphs.

---

> > ### Comment · Reviewer_RrLb · 2024-11-27
> >
> > Thanks for your response. However, I remain a main concern:  a GNN with low training efficiency and scalability seems to be contradictory. Besides, I would appreciate it if the authors provided experiments on heterophilic graphs.

---

> ### Comment · Area_Chair_XrRX · 2024-11-26
>
> Please check if the authors' response addresses your concerns.

---

> ### Author Response · Authors · 2024-11-27
> **Follow-Up to Reviewer RrLb**
>
> Thanks for reading through our rebuttal and providing additional feedback; please let us know if further details are needed.
>
> **Comment:**
> *I remain a main concern: a GNN with low training efficiency and scalability seems to be contradictory.*
>
> **Response:**
> A couple of key points are worth clarifying here.  First, the GAT and Sage baselines we compare against in Table 3 are based on highly optimized DGL implementations equipped with very efficient neighbor sampling (even GAT with appropriate NS can be quite efficient).  While not the absolute fastest among all possible GNNs, these baselines are extremely competitive, and our MuseGNN is still comparable with them.  Secondly, scalability itself is not equivalent to efficiency; the former emphasizes the ability to run on the largest graphs (e.g., without running OOM), but this need not occur at the absolute fastest speed to maintain scalability.
>
>
> **Comment:**
> *Besides, I would appreciate it if the authors provided experiments on heterophilic graphs.*
>
> **Response:**
> Our paper is entirely devoted to scaling UGNN models to the largest graphs, but unfortunately we are simply not aware of large-scale heterophily benchmarks.  In fact, the commonly-used heterophily graphs used for benchmarking are *tiny*.  For example, please see
> * *Graph Neural Networks with Heterophily*, AAAI 2021 (largest heterophily graph has **5201 nodes**).
> * *Beyond Homophily in Graph Neural Networks: Current Limitations and Effective Designs*, NeurIPS 2020 (largest heterophily graph has **7600 nodes**).
> * *A Critical Look at the Evaluation of GNNs under Heterophily: Are We Really Making Progress?*, ICLR 2023 (largest heterophily graph has **48921 nodes**).
>
> These graph benchmarks are all much smaller than even obgn-arxiv, and sampling-based models like MuseGNN are not optimal nor advisable in such cases.  Instead, to do well on these tiny datasets *full-graph* UGNN training can be directly applied, but such experiments have already been conducted in prior work (e.g., Yang et al., 2021), and hence are outside of our scope.
>
> We conclude by noting that existing papers devoted to GNN scalability like ours do not include heterophily benchmarks (see for example FreshGNN, MariusGNN, and LazyGNN papers and many references therein).  The only possible exception that we are aware of is SGFormer; however, we stress that this model *is only applied to heterophily benchmarks in a full-graph training manner* to compare with other graph transformers (not to be confused with GATs), an expensive model class often used with small graphs.  But this use case is exactly the same as prior UGNN work with full-graph training on heterophily graphs, and not relevant to large-graph scalability (our focus).  We hope these details help to alleviate concerns related to the inclusion of heterophily graphs.

---

> ### Comment · Reviewer_RrLb · 2024-12-02
>
> I'm not satisfied with this response. The Ogbn-arxiv, with around 170k nodes, is *not significantly* larger than heterophilic graphs that have around 50k nodes. Additionally, heterophilic graphs are receiving increasing attention in the field. The previous works did not focus on them, but that does not mean current work can ignore this issue. Besides, as far as I know, SGFormer is the most recent work in your list, so it is not surprising that it is the only one focused on heterophilic graphs.

---

> ### Author Response · Authors · 2024-12-02
> **Follow-Up to Reviewer RrLb**
>
> Thanks for continuing the discussion of our paper and allowing us another chance to respond.
>
> **Comment:**
> *The Ogbn-arxiv, with around 170k nodes, is not significantly larger than heterophilic graphs that have around 50k nodes.*
>
> **Response:**
> We agree that ogbn-arxiv is also a small graph, and good performance at such scales is not our purpose as simple full-graph training is feasible.  In fact, we only include ogbn-arxiv because of its widespread usage and as a sanity check to ensure our fixed MuseGNN architecture has consistent performance across different scales (see e.g., other scalability papers like FreshGNN, LazyGNN, etc. which all include ogbn-arxiv by convention).  Note also that graphs with 50k or fewer nodes are not relevant to sampling-based models like MuseGNN, where *each sampled subgraph* from a dataset like ogbn-papers100M already has around 65k nodes, larger than a full-graph of 50k nodes.
>
> **Comment:**
> *Heterophilic graphs are receiving increasing attention in the field. The previous works did not focus on them, but that does not mean current work can ignore this issue.*
>
> **Response:**
> We agree that heterophily graphs are receiving increased attention, but this attention so far is not primarily related to scalability, as current heterophily benchmarks can easily be handled with full-graph training and don't require scalability measures.  Rather, in the currently-available small-scale heterophily regime with full-graph training, UGNNs have already been thoroughly tested; two such examples are (Zheng et al., 2024) and (Yang et al., 2021).  There is not much more for us to show, since without its unique integration with sampling, MuseGNN offers no distinction from such existing UGNN approaches, at least until large-scale alternative benchmarks become available.  In the meantime, there is ample work to be done scaling GNNs in the homophily regime, e.g., the very recent VLDB 2024 proceedings from a couple months ago, including FreshGNN and the OUTRE model from Sheng et al.
>
> **Comment:**
> *SGFormer is the most recent work in your list, so it is not surprising that it is the only one focused on heterophilic graphs.*
>
> **Response:**
> We politely remark that SGFormer is not the most recent work we compare against (for example, FreshGNN is from VLDB in August 2024, while SGFormer is from last year's NeurIPS).  And we only chose to compare with SGFormer because they include results using sampling on papers100M, not because of small-scale full-graph heterophily cases.
>
> Perhaps it is also helpful to make a distinction between scaling typical GNNs versus scaling graph transformer models more narrowly.  SGFormer is targeting the latter, where the goal is to streamline global all-pair attention computations, which can sometimes be useful for addressing heterophily but is normally expensive even on tiny graphs.  This is surely why SGFormer includes such experimentation with tiny graphs, but as our scope is not scaling graph transformers, we do not.

---

### Official Review · Reviewer_y5VQ · 2024-11-04

**Soundness:** 3
**Presentation:** 3
**Contribution:** 3
**Rating:** 8
**Confidence:** 3

**Summary:**

The paper proposes a GNN model  that scales effectively to large datasets by incorporating sampled subgraphs into its energy function design. This approach allows the model to handle graphs with around 100 million nodes and high-dimensional features, achieving competitive accuracy on benchmarks exceeding 1TB in size. Additionally, it maintains desirable inductive biases and convergence guarantees.

**Strengths:**

1). The organization and writing of this paper are excellent.

2). The model enhances UGNN framework by incorporating sampling into its energy function design and demonstrates solid convergence properties.

3).  Empirical results suggest that MuseGNN maintains competitive accuracy and scalability across various task sizes, performing well on large node classification datasets exceeding 1TB.

**Weaknesses:**

1). How to determine \alpha and \lambda for different datasets?

2). Does the model's improvement differ for homogeneous and heterogeneous graphs? Can you provide some additional explanation?

**Questions:**

Refer to the content in the Weaknesses.

---

> ### Author Response · Authors · 2024-11-18
> **Response to Reviewer y5VQ**
>
> We appreciate the positive assessment and comments regarding the writing, convergence properties and experiments of our paper.  We address each constructive comment in turn below.
>
> **Comment:**
> *\[Weakness 1\] How to determine $\alpha$ and $\lambda$ for different datasets?*
>
> **Response:**
> Appendix A includes a paragraph describing the strategy for tuning hyperparameters.  Basically, we can simply tune them on down-sampled graphs and then apply to the originals (this dramatically reduces the cost of working with huge benchmarks).  Such a procedure is reasonable because of the relative stability of node-feature and network effects across scales.  Addtionally, our analysis from Theorem 5.2 further simplifies tuning by establishing an acceptable range of $\alpha$ for any choice of $\lambda$, i.e., because of this dependency there is more-or-less only a single degree-of-freedom to be tuned.  As a side note, it is also possible in principle to train both $\alpha$ and $\lambda$, along with other parameters $W$ and $\theta$, to avoid tuning altogether.  However, we have not carefully explored this strategy as merely testing a few different values of $\lambda$ was already sufficient to achieve good performance.  We can add further discussion of this possibility to Appendix A for reference.
>
> **Comment:**
> *\[Weakness 2\] Does the model's improvement differ for homogeneous and heterogeneous graphs? Can you provide some additional explanation?*
>
> **Response:**
> Good question.  While MuseGNN was primarily designed for scaling the most common UGNNs on homogeneous graphs, there does exist limited prior UGNN modeling work based on energy functions sensitive to heterogeneous graph structure. The basic idea is to update the trace term in Equation (4) to include additional trainable weight matrices that serve to align embeddings of nodes of different types and relationships. The HALO architecture (Ahn et al., 2022) is a full-graph example of this. But there is nothing to prevent us from extending the core techniques and analysis that undergird MuseGNN to scale such heterogeneous cases.  Doing so represents a useful direction for future work.

---

### Meta-Review · Area_Chair_XrRX · 2024-12-20

**Metareview:**

The manuscript proposes a graphical neural network training strategy based on minimizing a sub-sampled graphs into the energy minimization. Empirical results suggest the effectiveness and scalability of the training strategy. The reviewers largely agree that the approach is novel and interesting. Majority of the concerns by the reviewers have been addressed during the discussion. Thus the metareviewer recommends the paper for acceptance.

**Additional Comments On Reviewer Discussion:**

While one of the reviewers is not responsive during the discussion phase, other reviewers agree that the authors have addressed their main concerns.

---

### Decision · Program_Chairs · 2025-01-22

Accept (Poster)